# Balancing the film strain of organic semiconductors for ultrastable organic transistors with a five-year lifetime

Xiaosong Chen[1,8], Zhongwu Wang [1,2,8], Jiannan Qi[1,8], Yongxu Hu[1], Yinan Huang[1], Shougang Sun[1], Yajing Sun[1], Wenbin Gong[3], Langli Luo [4], Lifeng Zhang[4], Haiyan Du[5], Xiaoxia Hu[5], Cheng Han [2], Jie Li[1], Deyang Ji [1], Liqiang Li [1,6,7 ✉] & Wenping Hu [1,7]

The instability of organic field-effect transistors (OFETs) is one key obstacle to practical application and is closely related to the unstable aggregate state of organic semiconductors (OSCs). However, the underlying reason for this instability remains unclear, and no effective solution has been developed. Herein, we find that the intrinsic tensile and compressive strains that exist in OSC films are the key origins for aggregate state instability and device degradation. We further report a strain balance strategy to stabilize the aggregate state by regulating film thickness, which is based on the unique transition from tensile strain to compressive strain with increasing film thickness. Consequently, a strain-free and ultrastable OSC film is obtained by regulating the film thickness, with which an ultrastable OFET with a five-year lifetime is realized. This work provides a deeper understanding of and a solution to the instability of OFETs and sheds light on their industrialization.

[1] Tianjin Key Laboratory of Molecular Optoelectronic Sciences, Department of Chemistry, Institute of Molecular Aggregation Science, Tianjin University, Tianjin 300072, China. [2] SZU-NUS Collaborative Innovation Center for Optoelectronic Science & Technology, International Collaborative Laboratory of 2D Materials for Optoelectronics Science and Technology of Ministry of Education, Institute of Microscale Optoelectronics, Shenzhen University, Shenzhen 518060, China. [3] School of Physics and Energy, Xuzhou University of Technology, Xuzhou 221018, China. [4] Institute of Molecular Plus, Tianjin University, Tianjin 300072, China. [5] Analysis and Testing Center of Tianjin University, Tianjin University, Tianjin 300072, China. [6] Suzhou Institute of Nano-Tech and Nano-Bionics, Chinese Academy of Sciences, Suzhou 215123, China. [7] Joint School of National University of Singapore and Tianjin University, International Campus of Tianjin University, Fuzhou 350207, China. [8]These authors contributed equally: Xiaosong Chen, Zhongwu Wang, Jiannan Qi.
✉email: lilq@tju.edu.cn

Organic field-effect transistors (OFETs) are promising candidates in low-cost, flexible, and large-area electronics[1–4], but they have long suffered from stability-related problems. Although a few strategies have been developed to improve the stability of OFETs, including molecular design[5], interface engineering[6], and encapsulation[7], the shelf life of OFETs cannot meet the requirements of practical applications[7–14]. It is known that the aggregate state instability (morphology change and crystalline phase transition) of OSCs is one of the most important drivers of device failure among OFETs[15–19]. However, the underlying mechanism for aggregate state instability is still unclear, directly resulting in the lack of methods to realize an ultrastable aggregate state.

Strain, mainly stemming from the residual stress at the interface and boundary[20], is a key source of instability among conventional materials and devices, and thus, strain compensation or balance strategy is developed to improve the stability[20–25]. In the field of organic electronics, there are some reports about the relationship between the charge transporting property and the introduction of extrinsic strain[26–28], but the effect of the intrinsic film strain on device stability, as well as the strain balance strategy, in OSC films has never been reported until now.

In this work, we demonstrate that the OSC film undergoes a unique transition from tensile strain to compressive strain with increasing film thickness, which yields a strain-free and ultrastable OSC film by regulating the film thickness to ~200 nm. In films that are too thin (several to tens of nanometres), tensile strain leads to a prominent dewetting problem, while films that are too thick (above 300 nm) are prone to cracking due to compressive strain. Based on this finding, ultrastable and high-performance OFETs with a five-year lifetime are realized with a 200 nm dinaphtho[2,3-b:2′,3-f]thieno[3,2-b]thiophene (DNTT) film. Moreover, even after five years of storage, the 200 nm film device still exhibits excellent operational stability (10,000 s bias stress and 32,500 cycles test) and can steadily drive organic light-emitting diodes (OLEDs) for a long time, which strongly demonstrates its potential in practical applications.

## Results

**Thickness-dependent strain in OSC films.** It is well known that tensile and compressive strains are generally created in conventional inorganic polycrystalline films due to lattice mismatch, the difference in the coefficient of thermal expansion between the film and the substrate, and structural defects in the bulk of films[29–34]. These strains are detrimental to the structural stability of the film and device reliability. It has been reported that strain shows a strong dependence on film thickness[35–38]. In OSC films, the above drivers of strain, especially structural defects, are ubiquitous, so there should be a prominent strain. Furthermore, compared to covalently bonded inorganic semiconductors, OSCs have relatively weak intermolecular interactions, so the aggregate state stability should be more susceptible to film strain[26]. However, the effect of strain on OFET stability has yet to be fully elucidated, mainly due to the lack of understanding of strain in OSC films, directly leading to the lack of a strain balance strategy to stabilize the aggregate state of OSC films.

To systematically investigate the thickness-dependent strain in the OSC film, several DNTT films with different thicknesses were prepared under identical conditions (Fig. 1a). XRD measurements were carried out on these DNTT films and DNTT single crystals to confirm the existence of strain. XRD is a typical method to test the existence of strain, which has been widely utilized in various thin-film fields, including inorganic semiconductors[39], organic semiconductors[40], perovskites[20,25], and 2D materials[41] (detailed analysis shown in Supplementary Information section 1). For polycrystalline materials, strain is inevitable and can be caused by growth conditions and internal structural defects from the growth process, such as atomic interstitials, dislocations, and vacancies. Therefore, the lattice parameters of crystalline materials ($d$-spacing) are variable due to the existence of strain. As shown in Fig. 1b, the diffraction peak of the (001) crystal plane in the as-prepared DNTT films positively shifts with increasing film thickness. According to the Bragg's law, the interplanar spacing of the (001) crystal plane ($d_{001}$) is 1.6194 nm in the DNTT single crystal (Fig. 1b, purple dotted line). Taking the DNTT single crystal as the strain-free state, as shown in Fig. 1d, the relative strain of DNTT films (blue balls) with different thicknesses is calculated by Eq. 1:

$$\varepsilon = \frac{d - d_0}{d_0} \qquad (1)$$

where $\varepsilon$ is the relative strain in the DNTT films. $d_0$ and $d$ represent the interplanar spacing of the DNTT single crystal and films with different thicknesses, respectively.

Surprisingly, it can be found that the DNTT films undergo a unique transition from tensile to compressive state with increasing film thickness (Fig. 1d). By regulating the film thickness to an optimal range (~200 nm), the total strain is minimized to a nearly strain-free state close to a single crystal, while the thinner and thicker films suffer from tensile and compressive strain (Fig. 1e), respectively. The existence of strain in DNTT films was also demonstrated by Williamson–Hall analysis. Furthermore, we conducted XRD analysis with peak differentiation and fitting to separate the thin-film phase (FP) and bulk phase (BP) and found that the diffraction peaks of both FP and BP shifted with film thickness, which effectively excluded the influence of polymorphism (details shown in Supplementary Information section 1). As a result, the existing strain will lead to metastable states and even failure of OSC films. This unique strain behaviour and simple strain balance strategy have never been reported in OSC films. In inorganic materials, the strain balance or compensation strategy generally requires the elaborate material selection and structural design, such as using a substrate with a similar coefficient of thermal expansion[42], introducing an additional layer to compensate for the residual strain[25], and being passively compensated by external stimuli[43]. Although these strategies are effective in diminishing the residual strain in some cases, they also affect device performance and increase the device sophistication[20,25,42,43]. Our "strain balance" strategy of regulating film thickness provides an efficient and simple method for achieving a nearly strain-free OSC film.

To test the effect of thickness-dependent strain on the aggregate state of the OSC film, the morphology of OSC films with different thicknesses was monitored during storage. As expected, the morphology evolution of DNTT films with different thicknesses showed a considerable difference. As shown in Fig. 2a, b, and Supplementary Fig. 4a, b, the 20 nm DNTT films underwent a significant morphology evolution (i.e., dewetting process) during long-term storage, which is reported in previous work[44]. The molecules at the grain boundary, especially in the thinner films, could be observed to carry out mass transport, which leads to the aggregation of crystal grains, the formation of many holes and discontinuous networks, and an increase in the height of the crystal grains during the dewetting process (Fig. 2b). On the other hand, in the 300 nm DNTT film, although the films have stable morphology in the microscopic regime (Supplementary Fig. 5), numerous macroscale cracks formed in the whole film after being stored for approximately four months (Fig. 2e, f). In striking contrast, in the nearly strain-free thickness region, the 200 nm DNTT film exhibited excellent morphology stability, in which the crystal grains only slightly aggregated together without

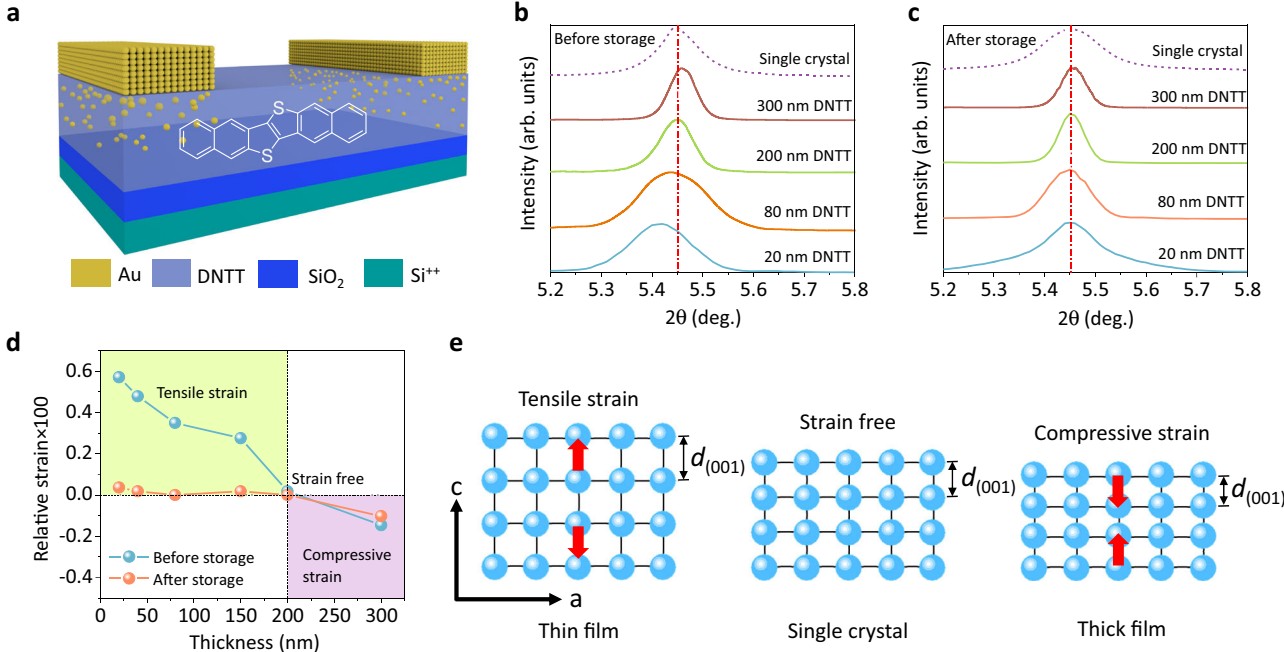

**Fig. 1 Model OFETs with different film thicknesses and strains. a** Schematic diagram of the OFET with an optimal OSC film thickness. The penetration of gold nanoclusters into the semiconductor layer is shown. XRD diffraction peaks of the (001) crystal plane of DNTT **b** before and **c** after five years of storage. **d** Relative strain of DNTT films with different thicknesses before and after storage. **e** Schematic diagram of the OSC lattice under tensile and compressive strain.

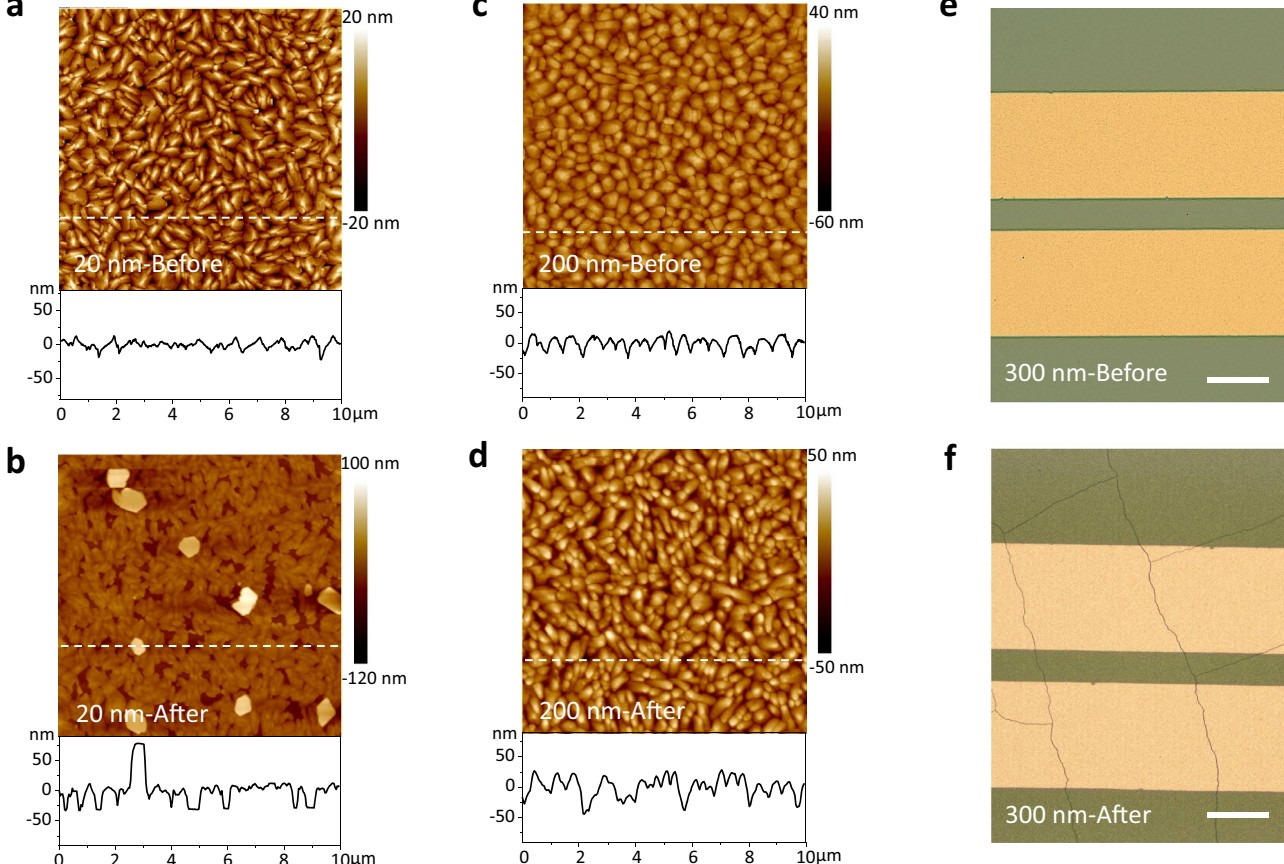

**Fig. 2 Aggregate state stability of DNTT films. a–d** Atomic force microscope images and section profiles of the 20 and 200 nm DNTT films before and after five years of storage. Optical images of 300 nm DNTT film **e** before and **f** after four months of storage by confocal laser scanning microscopy. The scale bar in **e** and **f** is 200 μm.

obvious change (Fig. 2c, d and Supplementary Fig. 4c, d). In fact, these films are fabricated and stored under identical conditions. XRD measurements on the DNTT films after storage showed that the film strain tends to release during storage (Fig. 1c, d), indicating that strain is the driving force for aggregate state instability.

**Interpretation of the thickness-dependent strain.** The unique thickness-dependent strain and aggregate state stability in the DNTT film can be understood as follows: it is reported that the growth behaviour in ultrathin films is mainly determined by the substrate properties such as the lattice parameter misfit, the mismatch of the coefficient of thermal expansion, and the mismatch of the surface energy[31,32,45–47]. The lattice parameters of DNTT crystals are heavily influenced by the substrate, especially in the first few monolayers[48]. In addition, the coefficient of thermal expansion of the $SiO_2$ dielectric layer ($\sim 10^{-6}$/°C)[49] is much lower than that of the OSC film ($\sim 10^{-4}$/°C)[50], which causes a mismatch of thermal stress, especially in an environment with a large temperature difference. These factors and the molecular conformation-induced stress reported in our previous work[51] synergistically form interface stress and thus lead to tensile strain in thin DNTT films, which serve as the driving force of the aggregate state instability. Nagarajan et al. revealed that the misfit strain is closely related to film thickness in thin-film systems[36] and is calculated by Eq. 2:

$$\varepsilon(h) = 1 - \frac{1-\varepsilon_0}{1-\varepsilon_0(1-h_c/h)} \quad (2)$$

where $\varepsilon(h)$ is the misfit strain caused by interface stress when the film thickness is $h$. $\varepsilon_0$ is the pseudomorphic misfit strain, which is a constant and is determined by the lattice parameters of the substrate and film. $h_c$ is the critical thickness for dislocation formation. With increasing film thickness, the misfit strain decreases monotonically, which agrees qualitatively with the tendency in Fig. 1d. By increasing the film thickness, the intermolecular interaction within the OSC film overshadows the constraint effect of interface stress, leading to a decrease in the tensile strain. Therefore, the thicker film has a lower possibility of dewetting[36,46,52,53]. As presented in Fig. 2c, d, the crystal grains of the 200 nm DNTT film only slightly aggregated together.

However, this does not mean that once the film thickness is higher than a critical value, the strain would constantly remain small or even zero. In fact, when the thickness of the DNTT films is much over 200 nm, the interface stress has a weak influence on the aggregate state of the OSC film. On the other hand, in the crystallization process during film formation, the volume shrinkage of OSCs and structural defects such as crystal boundaries produce and accumulate internal stress in the film[54–58], which induces a considerable compressive strain. Malzbender et al. systematically analyzed the internal stress related to film thickness, as depicted in Eq. 3[35] :

$$\sigma = \frac{E_f}{A}h^2 \quad (3)$$

where $\sigma$, $E_f$, $A$, and $h$ are the internal stress, elastic modulus, optical parameter, and thickness of the film, respectively. According to Eq. 3, the internal stress is proportional to the square of the film thickness, which implies that the excess internal stress will cause a severe film strain in the too-thick OSC film. When the thickness exceeds a critical range, the accumulated internal stress will be released by forming cracks (Fig. 2f).

**Effect of thickness-dependent strain on OFET stability.** The above discussion demonstrates that the thickness-dependent

strain results in the corresponding thickness-dependent aggregate state stability, which is closely related to the device stability. To confirm this point, several batches of DNTT OFETs with different thicknesses (20, 40, 80, 150, 200, and 300 nm) were prepared ("Experimental" section) under identical conditions, stored in the ambient environment, and periodically measured during five years of storage. Notably, the OFET with a 200 nm DNTT film presents excellent stability. As shown in Fig. 3a, the transfer curve of the 200 nm DNTT device after five years of storage has a similar or even better performance compared to that of the same device when it was newly prepared. In addition, the current on/off ratio and threshold voltage exhibit a smaller variation compared to those of OFETs with thinner and thicker films (Supplementary Figs. 6 and 7). For a facile comparison of the mobility stability (Supplementary Fig. 8), the normalized mobility ($\mu_d/\mu_0$, $\mu_d$, and $\mu_0$ denote the mobility at someday and the first day, respectively) is shown in Fig. 3b. The mobility of the 200 nm DNTT OFET exhibits the best stability with a slight variation of less than 10%, whereas the mobility of other thinner and thicker DNTT devices degrades dramatically after five years of storage.

Operational stability is another important indicator of OFET reliability. As shown in Fig. 3c, a constant bias over 10,000 s was applied to the 20 and 200 nm DNTT devices after five years of storage. The 200 nm DNTT device exhibits much better bias stress stability than the 20 nm DNTT device. In striking contrast, the drain current of the 200 and 20 nm DNTT devices remains at 90.7% and 2.3% of the initial value, respectively. Comparing their transfer curves before and after the bias stress test, the shift of threshold voltage ($\sim$5 V) of the 200 nm DNTT device was much smaller than that ($\sim$30 V) of the 20 nm DNTT device (Fig. 3d). Furthermore, after performing 32,500 cycles of testing (Fig. 3e), the 200 nm DNTT device still shows a normal switching function and negligible attenuation. Compared with the device stability in the literature (Table 1), the 200 nm DNTT device exhibits a much longer lifetime and retains a high performance of up to 90% of the initial value even after five years of storage. It is worth noting that these OFETs without an encapsulation layer were stored at different locations (Soochow, Shanghai, and Tianjin) with a wide range of temperatures (5–40 °C) and humidities (20–80%), indicating a high potential in the actual working environment.

To determine the origin of the above thickness-dependent OFET stability, the chemical and physical (aggregate state) aspects were tested. Traditionally, the origins of device instability were generally ascribed to chemical component degradation. DNTT was selected as the prototype OSC in this work to exclude the influence of chemical instability because of its large ionization potential and high chemical stability in an ambient environment[59]. To confirm this point, we characterized the UV–Vis absorption spectra of the DNTT film before and after storage (Supplementary Fig. 9). The negligible difference among absorption peaks demonstrates that the chemical structure of DNTT films remains stable when exposed to the ambient environment during storage[9,59], indicating that the chemical aspect is not the reason for the thickness-dependent OFET stability. Accordingly, the origins of thickness-dependent stability should stem from the aggregate state[16]. As stated above, the thinner and thicker films have tensile and compressive strain, which leads to dewetting and crack problems and thus accounts for the device instability. It is noted that an optimal thickness of OSC film can effectively balance the tensile and compressive strain to the strain-free state. Finally, excellent aggregate state stability and an ultrastable OFET were obtained. This "strain balance" strategy was also demonstrated to be effective in other OSC films (Supplementary Figs. 10, 11 and Section 6 in Supplementary Information), proving its high universality and potential in the OFET field.

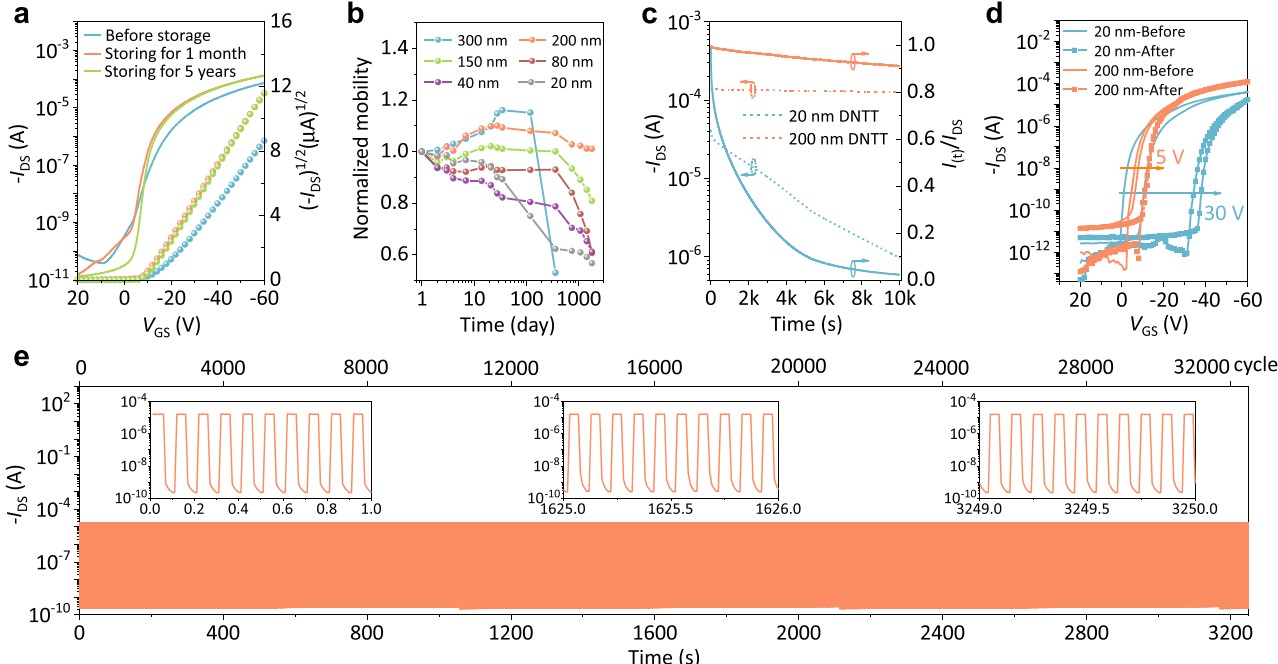

**Fig. 3 Effect of thickness-dependent strain on OFET stability. a** Transfer curves of the 200 nm DNTT OFET before storage and after storage for 1 month and five years. $V_{DS} = -60$ V. **b** Variation in the normalized mobility of OFETs with different thicknesses during the five-year storage period. **c** The bias stress effect tests of the 200 and 20 nm devices after five years of storage. $I(t)/I_{DS}$ (the solid line) denotes the ratio of the drain current at different times ($t$) to the initial drain current value ($I_{DS}$). $V_{DS} = -60$ V, $V_{GS} = -60$ V. **d** Transfer curves of the 200 and 20 nm DNTT OFETs before and after the bias stress test. 5 and 30 V represent the threshold voltage shift. $V_{DS} = -60$ V. The arrows denote the shift direction. **e** The 32,500 cycle switching tests of the 200 nm DNTT OFET after five years of storage (test frequency = 10 Hz, $V_{DS} = -30$ V, $V_{GS} = -30$ and 0 V, alternately). The channel length and width of the tested OFETs are 100 and 1000 μm, respectively.

**Table 1 The device stability of OFETs in the literature.**

| No. | Semiconductor | Strategy | Film thickness | Shelf life/operational stability | Degradation | Ref. |
|---|---|---|---|---|---|---|
| 1 | Pentacene | Encapsulation | 500 nm | 458 days[a, b] | 50% | 6 |
| 2 | DNTT | Ambient | 30 nm | 246 days[a] | 50% | 9 |
| 3 | TIPS-pentacene | Top-gate self-encapsulation | – | 210 days[a] | 7% | 10 |
| 4 | Ph-PXX | High ionization potential | 50 nm | 150 days[a] | 15% | 11 |
| 5 | TIPS-pentacene | Semiconductor/polymer blend | – | ~30 days[a] | <10% | 12 |
| 6 | TIPS-pentacene diF-TES-ADT | Top-gate self-encapsulation Microcrystal | 70 nm | ~5.9*10⁵ seconds[c] | ~5% | 13 |
| 7 | IDTBT | Top-gate self-encapsulation Water-free | - | ~25 hours[c] | ~5% | 14 |
| 8 | DNTT | Ps-buffered TPGDA dielectric | 70 nm | 6 months[d] | Bare shift | 15 |
| 9 | TIPS-pentacene | Self-encapsulation | 1 μm | 1400 hours | 75% | 66 |
| 10 | DNTT | PVS-buffered TPGDA dielectric | 70 nm | 8 months | 8.3% | 67 |
| 11 | DNTT | pIBA, pEGDMA, pPFDA dielectric | 50 nm | 140 days | >38% | 68 |
| 12 | DNTT | AlOₓ-SAM dielectric | 25 nm | 470 days | 20.8% | 69 |
| 13 | DNTT | Strain balance | 200 nm | 5 years[a] | 10% | This work |

[a]Shelf life.
[b]Theoretical estimation.
[c]Operational stability. The data are obtained from the literature.
[d]The lifetime was extracted from the same experimental conditions as our work.

**Self-optimized contact resistance**. Although the 200 nm film, which is approximately 10 times thicker than the usual OSC films in OFET, has a more stable aggregate state and thus is beneficial for realizing ultrastable devices, the huge contact resistance will seriously restrict its electrical performance, which is verified in the nonlinear characteristics in the output curves of the as-fabricated 200 nm DNTT devices (the linear region in the inset of Fig. 4a). Intriguingly, the output curves became linear after being stored for approximately one month (Fig. 4b). We periodically measured

the contact resistance and found that the contact resistance dropped rapidly by two orders of magnitude in the first 30 days (Fig. 4c and Supplementary Fig. 12). Subsequently, the contact resistance reaches a balanced state and remains the same level as the thin-film devices (Supplementary Fig. 13). It is known that the extraction of the exact mobility value in OFET is influenced by contact resistance. The gate voltage-dependent mobility of the 200 nm DNTT OFET before and after one month of storage is shown in Supplementary Fig. 14. After a one-month storage

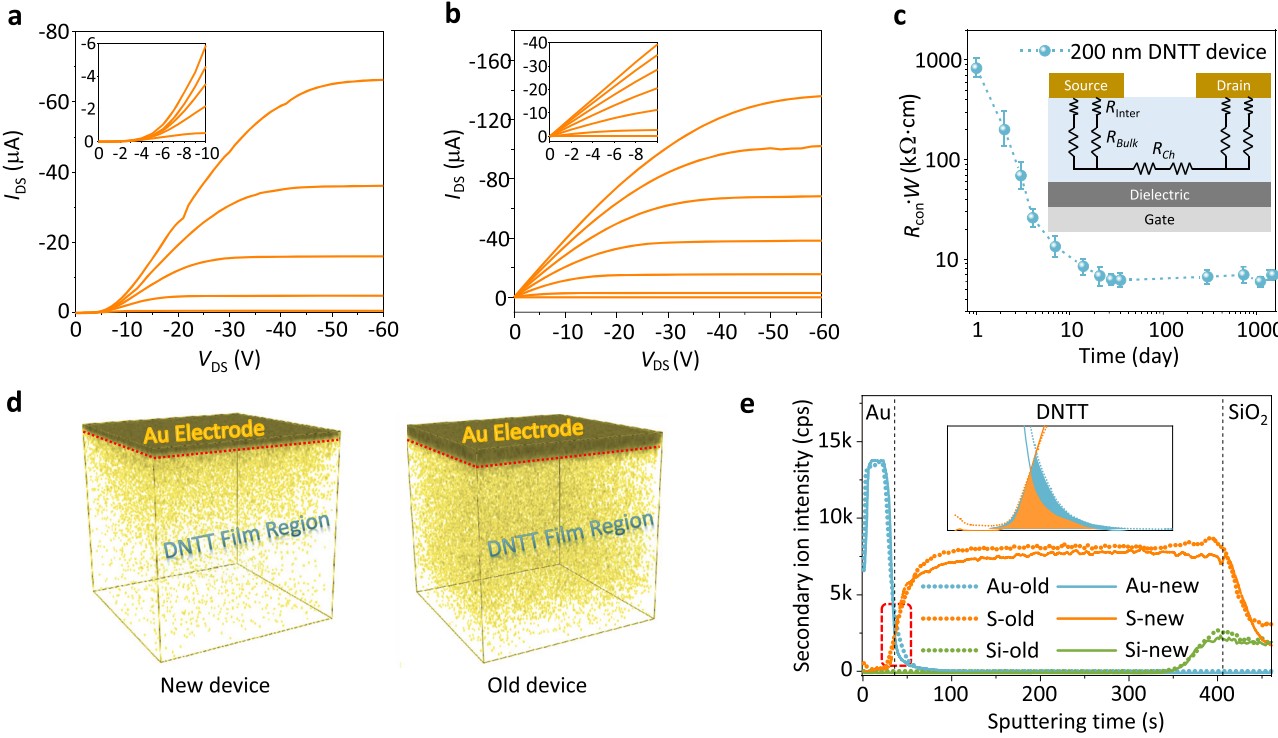

**Fig. 4 Self-optimized contact resistance.** Output curves of the 200 nm DNTT OFET **a** before and **b** after one month of storage. The insets are output curves in the linear regime. The channel length and width of the tested OFETs are 100 and 1000 μm, respectively. **c** The plot of contact resistance ($R_{con}$) versus storage time. "$W$" in the axis denotes the channel width. The inset is a schematic diagram of the charge transport circuit in OFET. $R_{Inter}$, $R_{Bulk}$, and $R_{Ch}$ are the electrode/semiconductor interface resistance, bulk resistance, and channel resistance, respectively. All error bars indicate standard errors based on multiple measurements. **d** The three-dimensional gold element distribution map (TOF-SIMS measurements) of (left panel) the new device and (right panel) the old device in the DNTT film region. **e** The depth profiles of the new device (solid line) and the old device (dashed line). The inset represents the enlarged overlap region of the gold element and DNTT, which is marked with a red rectangle in (**e**). The orange and blue regions indicate the overlap of the gold element and DNTT in the old device and new device, respectively.

period, the mobility shows a slighter gate voltage dependence due to the reduced contact resistance.

To clarify the underlying mechanism of the self-optimized contact resistance, we characterized the structure and components of the charge injection region (detailed information presented in Supplementary Figs. 15, 16). By time-of-flight secondary ion mass spectrometry (TOF-SIMS) measurement, the three-dimensional distribution of gold elements in the as-fabricated new device (left panel) and the old device after five years of storage (right panel) are presented in Fig. 4d. Combined with the etching time-dependent secondary ion intensity (Fig. 4e), the gold element of the old device exists at a longer etching time in comparison with the new device, proving a deeper penetration of gold nanoclusters (detailed explanation see Section 9 in Supplementary Information). The penetration of gold nanoclusters should account for the self-optimized contact resistance, which can be explained from the following two aspects: (i) the spontaneous diffusion of gold nanoclusters during storage could escape from the irradiation-damaged OSC region during thermally depositing electrodes and thus form better semiconductor/metal contact quality in the normal OSC region[60,61]. (ii) The penetrating gold nanoclusters could effectively reduce the contact energy barrier, which was demonstrated by the temperature-dependent I–V characteristics and the theoretical calculations in the Supplementary Information (Section 9, Supplementary 17, 18). Consequently, the huge contact resistance of the 200 nm DNTT OFET is significantly reduced, which demonstrates that it will not become an obstacle to practical applications.

To verify the effect of gold nanoclusters on the aggregate state of DNTT films, we deposited 3 nm gold nanoclusters on DNTT films and conducted the XRD and AFM measurements. Due to the high roughness of the DNTT surface, 3 nm gold nanoclusters are invisible in the AFM images. As shown in Supplementary Figs. 19 and 20, the morphology and XRD patterns of the DNTT film show negligible changes before and after the deposition of gold nanoclusters. Therefore, gold nanoclusters do not affect the aggregate states of organic semiconductors.

**Potential application of the ultrastable OFET.** The backplane circuit is one of the most important applications of field-effect transistors and requires high device stability. To demonstrate the excellent stability of the 200 nm DNTT OFET after five years of storage, we connected the 200 nm DNTT OFET with OLEDs in series. As depicted in Fig. 5, the 200 nm DNTT OFET exhibits a stable driven current and can steadily drive the OLED with no brightness decay for a long time. In contrast, the 20 nm DNTT device attenuates rapidly and is unable to normally drive the OLED in the end.

## Discussion

Finally, two points are noted below: (i) thick films and thin films have advantages and disadvantages. For instance, the thin film would be more flexible and transparent, but it also shows poor aggregate state stability. In contrast, the thick film would somehow reduce the transparency and might crack during storage. Therefore, it would be necessary to balance the aggregate state

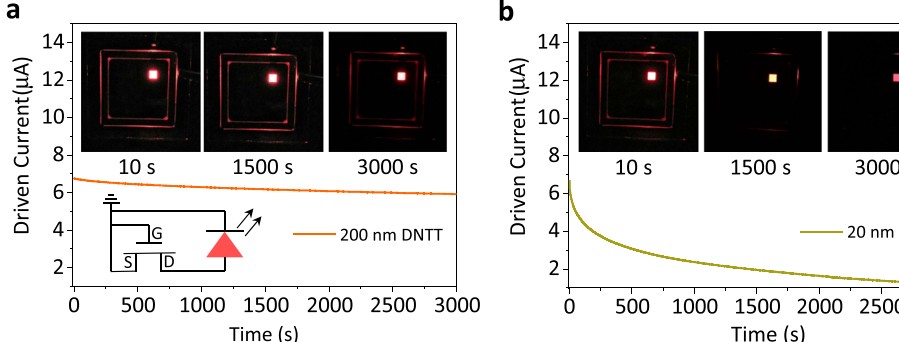

**Fig. 5 Potential application of the ultrastable OFET.** The driven current of **a** the 200 nm and **b** 20 nm DNTT OFETs versus time. The upper insets in **a** and **b** are pictures of the OLED driven by DNTT devices at different times. The bottom inset in **a** is the circuit diagram of the OLED driven by the DNTT OFET. G, S, and D indicate the gate electrode, source electrode, and drain electrode, respectively. The brightness of the OLED driven by the 200 nm DNTT OFET shows no visible change during the long-term operation, while it decays to a very weak value for the OLED driven by the 20 nm DNTT OFET. The bias was set as $V_{GS} = -80$ V and $V_{DS} = -4$ V.

stability, transparency, mechanical property, cost, and manufacturing difficulty for practical applications. In our work, an optimal film thickness is demonstrated, which well balances the film strain and enhances the aggregate state stability. Furthermore, the thick film may reduce the manufacturing difficulty of OSC film because the deposition of a uniform thick film over a large area has a relatively lower technique requirement compared to the thin film, especially in the solution process[62]. Adachi et al. pointed out that the thick film is of great interest to the industry with the requirements of high-yield, reliable fabrication, and thus intentionally utilized extraordinarily thick film to fabricate high-performance OLEDs by device engineering[63], strongly proving the advantage and potential of relatively thick film for device applications. (ii) Our work demonstrates that a proper thickness can produce a stable aggregate state. Therefore, the thinnest region in the film, including the grain boundary, would greatly influence the aggregate state stability. For a nonuniform film with a large roughness (such as DNTT and DPA films, Fig. 2a and Supplementary Fig. 11), the average thickness should be high enough to enable the grain boundary region to be thick enough to minimize the dewetting problem. However, if the film is uniform in thickness, it would be unnecessary to fabricate a very thick film to minimize the dewetting problem. For example, we deposited a 100 nm PDI-8CN2 film on the SiO₂ dielectric layer to fabricate OFET and stored it for approximately 1 year. The electrical performance and morphological evolution are presented in Supplementary Fig. 21. The PDI-8CN2 film has a relatively smooth surface and uniform thickness. After being stored for 332 days, the morphology of the 100 nm PDI-8CN2 film remains well and exhibits slight aggregation (Supplementary Fig. 21a, b). There is no obvious dewetting or cracking of the PDI-8CN2 film. The OFET based on the PDI-8CN2 film presented good stability after being stored for 332 days (Supplementary Fig. 21c).

In conclusion, stability is one of the most critical and unsettled issues for OFETs and lacks a simple solution for a long time. We found that the OFET instability stems from the residual strain in the OSC film and further reported an efficient strategy to balance the strain to significantly improve the device stability, which is based on the unique transition from tensile strain to compressive strain with increasing OSC film thickness. The tensile strain in an excessively thin film and the compressive strain in an excessively thick film lead to dewetting and crack problems, respectively. Significantly, this work realized an ultrastable OFET with a shelf life of over five years. This work builds a multidisciplinary bridge between organic electronics and material mechanics and would

exert a great and wide impact on the whole OFET field and other related organic optoelectronic devices.

## Methods

**Materials and Instruments.** DNTT and DPA were purchased from Aldrich. The electrical data were measured by a Keithley 4200 SCS and Semiconductor parameter analyzer PDA FS-Pro. AFM images were captured by a Bruker Dimension ICON-PT. XRD measurements in θ–2θ scan mode were carried out on a Rigaku MiniFlex600. The UV–Vis-NIR absorption spectra of the DNTT film deposited on quartz glass were measured with a SHZMADZU UV-3600 Plus spectrophotometer. TEM measurements were conducted on a Tecnai G2 F20 S-TWIN. TOF-SIMS measurements were conducted by TOF.SIMS5-100.

**Device fabrication and characterization.** The top-contact OFET devices were fabricated on silicon substrates. The substrates consist of a highly doped wafer (gate electrode) and a 300 nm thermally oxidized SiO₂ layer (dielectric layer). DNTT films with different thicknesses (20, 40, 80, 150, 200, and 300 nm) were deposited by vacuum thermal evaporation at 0.2 Å s⁻¹ under 10⁻⁴ Pa at a substrate temperature of 60 °C. Subsequently, gold electrodes were thermally deposited on OSC film in a vacuum at a rate of 0.5 Å s⁻¹. The channel length and width are 100 and 1000 μm, respectively. The OFET devices were stored at different locations (Soochow, Shanghai, and Tianjin) with a wide range of temperatures (5–40 °C) and humidities (20–80%).

Mobility was extracted by Eq. 4 at the saturation regime:

$$I_{DS} = \frac{\mu C_i W}{2L}(V_{GS} - V_T)^2 \tag{4}$$

Contact resistance was obtained by the transmission line method (TLM). According to the working mechanism of OFET, the total resistance at the on-state is made of contact resistance ($R_{con}$) and channel resistance ($R_{ch}$), which is described as Eq. 5:

$$R_{on} = \frac{\partial V_{DS}}{\partial I_{DS}} = R_{con} + R_{ch} = R_{con} + \frac{L}{W\mu C_i(V_{GS} - V_T)} \tag{5}$$

where $W$, $\mu$, $C_i$, and $V_T$ are the channel width, mobility, capacitance of the dielectric, and threshold voltage, respectively. By plotting the relationship of $R_{on}$ versus $L$ and reading the intercept of the curve on the ordinate axis, the contact resistance can be obtained.

In the demonstration application (i.e., OLED driven by the 200 nm DNTT OFET), the bias was set as $V_{GS} = -80$ V and $V_{DS} = -4$ V for ageing tests. The high gate voltage will cause a stronger bias stress and attenuate the driven current of the unstable OFET at a more rapid rate, which can simulate the long-term operational stability of the backplane.

**Theoretical calculation.** The electronic band structure calculation was performed based on density functional theory implemented in the Quantum Espresso package with the exchange−correlation functional of Perdew−Burke−Ernzerhof (PBE)[64], which usually underestimates band gap values by approximately 1 eV. However, this underestimation does not influence the interpretation of the results. The 400 eV plane wave basis cut-off was chosen, and the criteria of convergence for energy was set to be 1 × 10⁻⁵ eV. A more than 45 Å vacuum slab along the out-of-plane direction was added to eliminate the interaction between the periodic layers

(Supplementary Fig. 14). Grimme's dispersion corrections were also considered in our calculations[65].

## Data availability

The data presented in this study are available from the corresponding authors.

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

## Acknowledgements

The authors would like to thank the National Key Research and Development Programme (2018YFA0703200), National Natural Science Foundation of China (52073210, 62004128, 21905199, 21573277, 51633006, 62004138), Natural Science Foundation of Tianjin City (19JCZDJC37400, 194214030036), Beijing National Laboratory for Molecular Sciences (BNLMS202006), Key Research Programme of Frontier Sciences of Chinese Academy of Sciences (QYZDB−SSW−SLH031), and Fundamental Research Foundation of Shenzhen (JCYJ20190808152607389). China Postdoctoral Science Foundation (2021M702253). The authors sincerely appreciate the technological support from Vacuum Interconnected Nanotech Workstation (NANO-X), Suzhou Institute of Nano-Tech and Nano-Bionics (SINANO), Chinese Academy of Sciences, and Prof. Yi Cui, Dr. Xuefei Weng, Dr. Rong Huang, Gongzhong Miao, Zhiyun Li in NANO-X. The authors thank the Haihe Laboratory of Sustainable Chemical Transformations for financial support. We thank the facility center of the Institute of Molecular Plus at Tianjin University for the electron microscopy characterizations.

## Author contributions

X.C., Z.W., and J.Q. contributed equally to this work. X.C., Z.W., and L.L. conducted the experiments. X.C., Z.W., and L.L. wrote the manuscript. Y.S. conducted the theoretical calculation. W. G., L.L.L., and L.F.Z. assisted in the strain analysis. H.D. and X.H. provided professional guidance for the XRD measurements and data analysis. Y.H., Y.N.H., S.S., C.H., J.L., and D.J. assisted in the experiments. W.H. provided valuable suggestions on this work. L.L. conceived and supervised this work.

## Competing interests

The authors declare no competing interests.
