## [Peer Review File · Nature Communications]

Reviewer comments, initial review

Reviewer #1 (Remarks to the Author):

This manuscript reports the highly stable organic transistors based on DNTT. The stability is derived from the film thickness. The authors found that 200 nm is the balanced thickness without any strain. Thinner films suffer from tensile strain, while thicker films have more compression strain. This film thickness dependency was demonstrated in organic transistors. The shelf-storage stability of the device performances was evaluated for 5 years. The 200 nm thick device display any significant degradation, which suggests the bright future of practical applications. Overall, I think that this is a nice contribution. After suitable revision, it can be accepted.

1. How did the authors prepare films? Vacuum deposition or solution processes? If solution processes are adopted, annealing conditions should be provided.
2. Why did the authors select 5 years as a long term storage period? Detailed storage conditions should be provided.
3. AFM and confocal scanning microscopy images are shown in Fig. 2 and Supplementary Information. It is fair to show the confocal laser scanning microscopy images with the 200 micrometer scale for 20nm and 200nm thick samples.
4. It is not clear how to experimentally determine the parameters in equations (2) and (3).
5. In Table 1, there are many organic semiconductors. The reviewer is wondering if there is an optimum film thickness for each semiconductor or 200 nm is the universally optimized thickness for organic semiconductors.
6. Gold nanoclusters are formed at the surface of the organic semiconductor layer. The reviewer wonders if this affects the aggregate states of organic semiconductors. XRD and AFM measurements may give an insight.
7. There are some careless mistakes.

In page 4, ...that strain show...

In page 8, it is described that ...stored for about four months (Fig. 2e, 2f). However, in the caption of Fig. 2, it is described that they are five year storage samples.

In page 18, Prof. Adachi et al. (remove "Prof.")

Reviewer #2 (Remarks to the Author):

This is the third time I get to review this manuscript, and since the current version is very similar to the previous versions, I can essentially repeat what I wrote earlier:

Introduction:

The authors claim that "it is known" that organic transistors "fail" because of "aggregate state instability (morphology change and crystalline phase transition)". However, this claim is not supported by the references cited here (references 15-17). Reference 15 reported that "DNTT OTFTs showed exceptionally good environmental stability". Reference 16 reported that monolayers of DNTT were observed to break up, and it was speculated that this might occur due to mechanical strain, but reference 16 also demonstrated that this break-up does not occur when the films have a thickness of more than one monolayer. In reference 17, it is stated that "strain ... caused by the structural misfit ... cannot serve as an explanation for the dewetting in the present case, so that the reason remains unclear." The authors' claim regarding the failure of organic transistors thus remains hypothetical.

The authors claim that organic-semiconductor films with a small thickness often suffer from dewetting caused by tensile strain. The dewetting of thin molecular films is indeed well documented in the literature, but whether this dewetting is caused by mechanical strain or by some other mechanism is not clear at all, as explained in reference 17.

The fact that the dewetting of monolayer-thick DNTT films can be alleviated by increasing the thickness of the DNTT films has been reported previously (reference 16).

Section "Thickness-dependent strain in OSC film":

The analysis of the mechanical strain is rudimentary and likely flawed. The authors have only calculated (using Equation 1) the relative difference between the (001) d-spacing of the DNTT films (measured by x-ray diffraction) and that of a DNTT crystal and then assumed that this relative difference in d-spacings represents the mechanical strain in the DNTT films. However, the observed d-spacings are more likely the result of polymorphism, rather than strain, since vacuum-deposited films of small-molecule semiconductors, such as DNTT, are often characterized by a coexistence of various phases with different d-spacings (Appl. Phys. Lett., vol. 90, p. 081903, 2007; Phys. Rev. B, vol. 85, p. 195308, 2012; Adv. Funct. Mater., vol. 26, p. 2233, 2016; Adv. Mater., vol. 29, p. 1604833, 2017; ACS Appl. Mater. Interfaces, vol. 9, p. 11977, 2017). Since the authors have not performed a Williamson-Hall (W-H) analysis, their claims regarding the magnitude of the strain in the DNTT films are purely speculative.

Besides, if there was indeed significant strain in the DNTT films, it is highly unlikely that the strain would display a monotonic dependence on the film thickness (as the authors indicate in Figure 1d); it is far more likely that a transition from a surface layer to a bulk layer would occur beyond a critical film thickness (Adv. Funct. Mater., vol. 26, p. 2233, 2016).

The authors report that they did not observe any changes in the morphology of the nominally 200-nm-thick DNTT films over a period of five years using atomic force microscopy (Figure 2c,d), but this AFM analysis is meaningless, since AFM images show only the surface of the films, but not the morphology of the material buried underneath the surface.

Section "Interpretation of the thickness-dependent strain and aggregate state stability":

The authors claim that the lattice parameters of the DNTT films are different from the lattice parameter of the amorphous silicon dioxide gate dielectric. This makes no sense, because amorphous materials (such as silicon dioxide) obviously do not have any lattice parameters, which also means that there is no misfit and hence no misfit strain.

The authors claim that the thermal-expansion coefficient of DNTT is different from that of the silicon dioxide. This statement is meaningless, since the thermal-expansion coefficients of single-crystalline and polycrystalline organic semiconductors are usually highly anisotropic, and the thermal-expansion coefficient often has a positive value in one lattice direction, but a negative value in another lattice direction.

Section "Effect of thickness-dependent strain on OFET stability":

In Table 1, the authors ignore all previous publications in which the excellent long-term stability of DNTT transistors has been reported (reference 15; Org. Electronics, vol. 15, p. 1998, 2014; Adv. Electron. Mater., vol. 2, p. 1500209, 2016; Org. Electronics, vol. 61, p. 65, 2018).

Section "Self-optimized contact resistance by the spontaneous penetration of gold nanoclusters":

The contact resistance of the DNTT transistors reported here (even for a nominal DNTT thickness of 20 nm) is about an order of magnitude larger than what has been reported in the literature for DNTT transistors; this indicates that there are problems with the device fabrication process employed here.

The observation that the contact resistance of organic TFTs decreases over time has also been reported (see, for example, Electronic Mater. Lett., vol. 15, p. 166, 2019; Phys. Rev. B, vol. 76, p. 184112, 2007).

The fact that this improvement in contact resistance may be related to the penetration of gold clusters into the semiconductor film has also been known for a long time (see, for example, Adv. Mater., vol. 14, p. 961, 2002; J. Appl. Phys., vol. 93, p. 5201, 2003; Appl. Phys. A, vol. 95, p. 225, 2009; Org. Electronics, vol. 15, p. 1738, 2014; Org. Electronics, vol. 69, p. 92, 2019; Jpn. J. Appl. Phys., vol. 58, p. SIIB28, 2019; ACS Appl. Mater. Interfaces, vol. 12, p. 30616, 2020).

For these reasons, I am unable to recommend publication in Nature Communications.

Reviewer #3 (Remarks to the Author):

It is well-known that stability is one of the most important and challenging issues in the field of organic electronics. In "Balancing the Film Strain of Organic Semiconductor for Ultra-stable Organic Transistors with a Five Years Lifetime", the authors report a stable organic field-effect transistor (OFET) with 5 years of lifetime by a strain balance strategy, and disclose that the intrinsic tensile and compressive strain existing in the organic semiconductor (OSC) film are the key origins of the aggregate state instability and device degradation. To the best of my knowledge, this truly impressive 5-year lifetime is a record result. The strain balance strategy based on thickness-dependent strain evolution in organic semiconductors opens up a new perspective to device stability in organic electronics. The results resolve one of the key challenges in the field of organic electronics in a unique and new way. The topic of this work is important and would be appealing for the broad readers of Nature Communications from both academia and industry. Therefore, I recommend publishing this manuscript in Nature Communications after minor revisions.

1. XRD is a typical method for strain analysis. The authors are suggested to discuss the method and the representative literature (Nat. Mater. 20, 1337 (2021); Nat. Mater. 20, 1337 (2021)).
2. In Fig. 2b, the film became discontinuous after storage. Where are the disappeared semiconductor molecules? The authors are suggested to explain this phenomenon.
3. In lines 140-142 on page 8, the authors state that "it is reported that the growth behavior in ultrathin films is mainly determined by the substrate properties such as the lattice parameter misfit and the mismatch of coefficient of thermal expansion". I think the mismatch of surface energy should also be taken into account.
4. In lines 218-219 on Page 12, "the 200 nm DNNT device exhibits a much longer lifetime and 218 remains high performance up to 90% even after five-year storage". It seems that the mobility after five-year storage is even slightly higher than that of the fresh device, as presented in Fig. 3b. Please explain it.
5. Supplementary Fig. 10 is not referenced in the manuscript. Please check and correct it.

Note: Fig. R*, Fig. *, and Supplementary Fig. * denote the figure in Reply, Maintext, and Supplementary Information, respectively.

Reply for Reviewer 1:

Reviewer #1 (Remarks to the Author):

Comment 1

This manuscript reports the highly stable organic transistors based on DNNT. The stability is derived from the film thickness. The authors found that 200 nm is the balanced thickness without any strain. Thinner films suffer from tensile strain, while thicker films have more compression strain. This film thickness dependency was demonstrated in organic transistors. The shelf-storage stability of the device performances was evaluated for 5 years. The 200 nm thick device display any significant degradation, which suggests the bright future of practical applications. Overall, I think that this is a nice contribution. After suitable revision, it can be accepted.

Our reply: We deeply appreciate your approval of the significance of our work. Your professional and constructive comments and suggestions guide us to think about some points more deeply and are very helpful to improve the quality of the manuscript. We will answer your concerns one by one below.

Comment 2

How did the authors prepare films? Vacuum deposition or solution processes? If solution processes are adopted, annealing conditions should be provided.

Our reply: Thanks for pointing out the unclear information! All the films used in the work were prepared by vacuum deposition. The detailed film preparation process is as follows: DNNT films with different thicknesses (20 nm, 40 nm, 80 nm, 150 nm, 200 nm, and 300 nm) were deposited by vacuum thermal evaporation at 0.2 \AA s^{-1} under 10^{-4} Pa at a substrate temperature of $60 \text{ }^\circ\text{C}$, respectively.

Our revision: The process details are supplemented with more information in the Method Section.

Comment 3

Why did the authors select 5 years as a long term storage period? Detailed storage conditions should be provided.

Our reply: Thanks for your valuable suggestions! Actually, our group has focused on the stability issue of OFETs since 2014. For the lifetime testing, we fabricated a batch of devices to track the performance in each research topic about the stability of OFETs. In fact, the thickness-dependent stability has been observed several years ago. To deeply understand the mechanism and to obtain a recorded shelf-life, we tracked the devices as long as possible, which has spanned 5 years. During this period, we have been continuing to study the underlying origin for this phenomenon, and until last year, we finally get the convincing evidence and explanation from the aspect of strain-induced instability of the aggregate state of organic semiconductors. This finding was inspired by our work on strain sensors¹ and strain-related stability of perovskite,^{2,3} but has never been reported in the field of organic transistors. Due to the long time span, the temperature and the humidity are not recorded for every test. However, the average temperature and humidity range in the past five years can be queried in historical weather. As already mentioned in the manuscript (Page 13), these OFET devices were stored at different locations (Soochow, Shanghai, and Tianjin) with a wide range of temperatures (5 °C~40 °C) and humidity (20%~80%).

Reference

- [1] Li, H. et al, Eggshell-inspired membrane—shell strategy for simultaneously improving the sensitivity and detection range of strain sensors. *Sci. China Mater.* 64, 717 (2021)
- [2] Xue, J. et al. Regulating strain in perovskite thin films through charge-transport layers. *Nat. Commun.* 7, 11156 (2020).
- [3] Liu, D. et al. Strain analysis and engineering in halide perovskite photovoltaics. *Nat. Mater.* 20, 1337 (2021).

Our revision: The detailed storage conditions are added in the Method section (Page 22).

Comment 4

AFM and confocal scanning microscopy images are shown in Fig. 2 and Supplementary Information. It is fair to show the confocal laser scanning microscopy images with the 200 micrometer scale for 20nm and 200nm thick samples.

Our reply: Thanks for your kind reminders. We supplemented the confocal laser scanning microscopy images with the 200 micrometer scale for 20 nm and 200 nm thick samples before and after storage for 5 years in Fig. R1. As shown in Fig. R1a and R1b, the 20 nm DNTT film occurs severe dewetting phenomenon during storage. The DNTT film (the blue region) aggregated together and the SiO₂ substrate (the dark dots) became visible (enlarged view in Fig. R1b). For 200 nm DNTT film (Fig. R1c and R1d), the morphology didn't show obvious change. The slight color difference of the same samples before and after storage comes from the change of film thickness and the condition of the light source in instruments. Because of the low resolution and magnification compared to AFM, it's difficult to clearly observe the grains in the confocal laser scanning microscopy images.

Fig. R1. The confocal laser scanning microscopy images of 20 nm DNTT films (a) before

and (b) after storing for five years. The confocal laser scanning microscopy images of 200 nm DNTT films (c) before and (d) after storing for five years.

Our revision: The confocal laser scanning microscopy images with the 200 micrometer scale for 20nm and 200nm thick samples before and after 5 years' storage are added in the Supplementary Fig. 4.

Comment 5

It is not clear how to experimentally determine the parameters in equations (2) and (3).

Our reply: Equations (2) and (3) are theoretical equations of thickness-dependent strain/stress. They can be used to **qualitatively rather than quantitatively** describe the relationship between strain/stress and film thickness. The parameters cannot be precisely measured in organic semiconductor film because of the high technical difficulty and the unique properties of organic semiconductors.

For equation (2),

$$\varepsilon(h) = 1 - \frac{1-\varepsilon_0}{1-\varepsilon_0(1-h_c/h)} \dots\dots\dots (2)$$

where $\varepsilon(h)$ is the misfit strain caused by interface stress when the film thickness is h . The film thickness can be accurately given by AFM images. ε_0 is the pseudomorphic misfit strain, which is usually regarded as a constant and is determined by the lattice parameters of substrate and film. h_c is the critical thickness for dislocation formation. The critical thickness criteria represent the film thickness in which the elastic energy release provided by the edge component of a misfit dislocation is equivalent to the energy increase due to the line energy of the dislocation. The total strain energy per unit area of a film partially relaxed by misfit dislocations is given as:¹

$$U^{total} = \frac{hE}{1-\nu} (\varepsilon_h - \rho|\mathbf{b}|\cos\lambda)^2 + \rho \frac{E|\mathbf{b}|^2}{4\pi(1+\nu)} \left(\frac{1-\nu\cos\beta^2}{1-\nu}\right) \ln\left(\frac{\alpha h}{|\mathbf{b}|}\right) \dots\dots\dots (R1)$$

where ε_h and ρ are the misfit strain and the misfit dislocation density given as the number of dislocations per unit length along one of the principal directions, $|\mathbf{b}|$ and λ are the magnitude of the Burgers vector and the angle between the Burgers vector and a line which

lies within the interface and in a plane normal to the dislocation line, β is the angle between the dislocation line and the Burgers vector, ν is Poisson's ratio, and ρ is the cutoff parameter used to describe the subcontinuum energy of the dislocation core. The critical thickness for the introduction of the first misfit dislocations is determined by the boundary condition:

$$\left(\frac{\partial U^{total}}{\partial \rho}\right)_{\rho=0} = 0$$

However, the parameters related to the Burgers vector in equation R1 are very difficult to determine, because organic semiconductor films are too weak to carry out high-resolution structure measurement such as TEM. However, it does not influence the validity of equation (2).

For equation (3),

$$\sigma = \frac{E_f}{A} h^2 \dots\dots\dots(3)$$

Where σ , E_f , A , and h are the internal stress, the elastic modulus, the optical parameter, and the thickness of the film, respectively. The elastic modulus is influenced by the film structure. For example, for organic semiconductor film with high-density micro-porous, the elastic modulus is much lower than that of dense film. The distribution of micro-porous in DNTT films is fully random. Besides, with the increasing thickness, the density of micro-porous in DNTT film varies with the depth. The unevenly distributed structure makes it difficult to determine the elastic modulus in polycrystalline DNTT films. According to the reported values in literature, the elastic modulus of DNTT films is about 2~3GPa.²

Reference

[1] Speck, J. S. & Pornpea W. Domain configurations due to multiple misfit relaxation mechanisms in epitaxial ferroelectric thin films. I. Theory. J. Appl. Phys. 76, 466 (1994).
 [2] Fukuda, K., et al. Free-Standing Organic Transistors and Circuits with Sub-Micron Thicknesses. Sci. Rep. 6, 27450 (2016).

Our revision: We explain the qualitative feature of equations (2) and (3) in the maintext.

Comment 6

In Table 1, there are many organic semiconductors. The reviewer is wondering if there is an optimum film thickness for each semiconductor or 200 nm is the universally optimized thickness for organic semiconductors.

Our reply: Thanks for your insightful comment!

(1) Our work demonstrates that a proper thickness can produce a stable aggregate state, and too thin film has the dewetting problem. Therefore, the thinnest region in film including grain boundary would greatly influence the aggregate state stability. As for a non-uniform film with big roughness (such as DNTT and DPA films, Fig. 2a and Supplementary Fig. 11), the average thickness should be high enough to enable the grain boundary region to be thick enough to minimize the dewetting problem. On the other hand, if the film is uniform in thickness, it would be unnecessary to fabricate a very thick film to minimize the dewetting problem. For example, we ever deposited 100 nm PDI-8CN2 film on the SiO₂ dielectric layer to fabricate OFET and stored it for about 1 year. The electrical performance and morphological evolution are presented in Fig. R2. The PDI-8CN2 film has a relatively smooth surface and uniform thickness. After storing for 332 days, the morphology of 100 nm PDI-8CN2 film remains well and occurs partially slight aggregation (Fig. R2a,b). There is no obvious dewetting or cracking of the PDI-8CN2 film. The OFET based on PDI-8CN2 film presented good stability after storing for 332 days (Fig. R2c).

Fig. R2. (a,b) The morphology and (c) transfer curves of PDI-8CN2 OFET before and after storing for 332 days.

(2) The strain in organic semiconductor films varies from material and substrate. The tensile

strain is mainly determined by the molecule/substrate interaction. To realize a balanced strain state, the film thickness has to be increased to a certain value to produce a comparable compressive strain. Thickness is only the apparent indicator in this strain balance strategy. Therefore, for different organic semiconductors and substrates, the optimum thickness needs to be explored according to the molecule/substrate interaction and intermolecular interaction.

Our revision: We have added the discussion about the optimum film thickness in the revised manuscript (Page 20 and Supplementary Fig. 21).

Comment 7

Gold nanoclusters are formed at the surface of the organic semiconductor layer. The reviewer wonders if this affects the aggregate states of organic semiconductors. XRD and AFM measurements may give an insight.

Our reply: Thanks for your professional comment. To verify the effect of gold nanoclusters on the aggregate state of DNTT films, we deposited 3 nm gold nanoclusters on DNTT films and characterized the XRD and AFM measurements. Due to the high roughness of DNTT surface, 3 nm gold nanoclusters are invisible in the AFM images. As shown in Fig. R3 and R4, the morphology and XRD patterns of DNTT film show negligible change before and after depositing Au nanoclusters. Therefore, gold nanoclusters don't affect the aggregate states of organic semiconductors.

Fig. R3. AFM images of DNTT film before and after depositing 3 nm gold nanoclusters.

Fig. R4. XRD patterns of DNTT film before and after depositing 3 nm gold nanoclusters.

Our revision: We have supplemented the effect of gold nanoclusters on the aggregate states of organic semiconductors in the revised manuscript (Page 18), and the AFM images and XRD patterns of DNTT film in the revised Supplementary Information (Supplementary Fig. 19, 20).

Comment 8

There are some careless mistakes.

In page 4, ...that strain show...

In page 8, it is described that ...stored for about four months (Fig. 2e, 2f). However, in the caption of Fig. 2, it is described that they are five year storage samples.

In page 18, Prof. Adachi et al. (remove "Prof.")

Our reply: Thanks for your keen observation and kind reminders!

Our revision: We have carefully revised the mistakes in the revised manuscript.

Reply for Reviewer 2:

Reviewer #2 (Remarks to the Author):

Comment 1

This is the third time I get to review this manuscript, and since the current version is very similar to the previous versions, I can essentially repeat what I wrote earlier:

Our reply: We appreciate you for your critical and constructive comments, because your comments truly guide us to think about some points deeply and improve our manuscript. In this round, we have carefully addressed all your comments, doubt, and denial. Here, we politely push back on some of your key comments that are the basis for rejecting our paper: (1) it is unconfirmed that organic transistors fail because of aggregate state instability (morphology change and crystalline phase transition); (2) the strain analysis is rudimentary and likely flawed, and the strain-induced instability is hypothetical; (3) our paper is lack of novelty because of the above points as well as the reported thickness-dependent film stability, stable DNTT devices, and the evolution of contact resistance.

For clarity and your convenience, we briefly list our reply for the above three points:

- (1) It is well-recognized that the unresolved instability problem is the most important bottleneck for the application of organic transistors. Generally, instability of organic devices is attributed to the chemical and physical aspects,¹⁻³ i.e., chemical degradation (such as photooxidation of pentacene) and physical change (referring to aggregate state change) (Figure R5). The chemical degradation has been well-resolved by developing chemically-stable organic semiconductors. The aggregate state change (film morphology change and phase transition) has been assigned to be the main reason for device instability, but it is just the apparent reason, and the underlying reason (or driving force) for the aggregate state change is not revealed, resulting in no effective solution (Figure R5). In our work, **we revealed that strain in the film is one of the most important factors for the**

aggregate state change, and developed a novel strain-balance way to realize the ultra-stable film and ultra-stable devices.

The origin of performance degradation in OFETs

Fig. R5. The origin analysis of performance degradation in OFETs

Reference

- [1] Henning S. Reliability of organic field-effect transistors. Adv. Mater. 38-39, 3859 (2009).
- [2] Jorgensen, M. et al. Stability of Polymer Solar Cells. Adv. Mater. 24, 580 (2012).
- [3] Aziz, H. & Popovic, Z.D. Degradation phenomena in small-molecule organic light-emitting devices. Chem. Mater. 16, 4522 (2004).

(2) According to your suggestions, we have carefully performed W-H analysis and other supporting experiments, which together with the related literature firmly confirm the existence of the strain and its key role on the instability.

(For details, see our reply for comments 3 and 5).

(3) The key novelty of this work is as follows: We firstly demonstrated that OFET instability mainly stemmed from the residual strain in OSC film, and further reported a novel and efficient strategy to balance the strain and significantly improve the device stability, which is based on the unique transition from tensile strain to compressive strain with the increase of OSC film thickness. Moreover, an ultra-stable organic transistor for **5 years** is realized with the strain balance strategy, which has never been reported before.

(a) As for the reported thickness-dependent film stability, **it is just a phenomenon observation, and the underlying mechanism (strain) has not been revealed before**. Furthermore, the transition from tensile strain to compressive strain and the strain-balance strategy have not been reported, which are essential for the understanding and solution of device instability (for details, see our reply for comment 4). (b) Compared to our devices prepared by the strain-balance strategy, **the reported stable DNTT devices show quite a shorter storage period**, which actually proves the significance of our work (for details, see our reply for comment 10). (c) As for the reported evolution of contact resistance, **it is just a technical, but not the key point of this work, which just proves thick film would not suffer from the large contact resistance** (for details, see our reply for comments 11 and 12).

Dear reviewer #2, we believe that you acknowledge the key role of device stability for practical applications. Despite its significance, the stability study far lags behind the development of materials and processes because it requires much time and resources and often yields unsatisfactory results (*Nat. Energy.* 5, 947 (2020)). Although some literature claim that their devices show good (or excellent) stability, the lifetime and related electrical parameters could not meet the requirements of practical applications. If the instability problem could not be resolved, the field of organic transistors would die away gradually.

Our work provides a novel understanding and solution to the instability problem, and the

unprecedented 5-year lifetime is already enough for some applications. Thus, this work would exert a significant effect on the field of organic transistors. In the previous review rounds, you may doubt whether there is strain in organic film and whether the strain plays a key role in device instability. In this round, we have carefully addressed all your concerns in a convincing way by the supplementary experiments, comparative data analysis, and supporting evidence in the literature. Please spare some time to read the following reply. We sincerely hope you can support this work, which would be a great encouragement for us to continue the stability study. Thanks a lot for your understanding and support.

Comment 2

The authors claim that “it is known” that organic transistors “fail” because of “aggregate state instability (morphology change and crystalline phase transition)”. However, this claim is not supported by the references cited here (references 15-17). Reference 15 reported that “DNTT OTFTs showed exceptionally good environmental stability”. Reference 16 reported that monolayers of DNTT were observed to break up, and it was speculated that this might occur due to mechanical strain, but reference 16 also demonstrated that this break-up does not occur when the films have a thickness of more than one monolayer. In reference 17, it is stated that “strain ... caused by the structural misfit ... cannot serve as an explanation for the dewetting in the present case, so that the reason remains unclear.” The authors’ claim regarding the failure of organic transistors thus remains hypothetical.

Our reply: Thanks for the helpful comments which guide us to make this point clearer. In fact, from the comprehensive aspects of stability, references 15-17 support that aggregate state instability is one of the main origins of device failure during long-term storage. The reasons are clarified as follows:

- (1) Generally, the instability of organic devices is attributed to the chemical and physical aspects¹⁻³, i.e., chemical degradation (such as photooxidation of pentacene) and physical change (referring to aggregate state change). The chemical degradation has been well-resolved by developing chemically-stable organic semiconductors. For example, DNTT shows excellent chemical stability, just as stated in Reference 15

("DNTT is chemically much more stable, so oxidative degradation of the molecule does not occur"). Therefore, **the electrical degradation of the device mainly stems from the aggregate state change (film morphology change and phase transition), which has been verified by many literatures including ref. 16-17.** As shown in Fig. R6, "owing to this excellent chemical stability, the changes in the measured electrical properties of the ultrathin DNTT films can be unequivocally assigned to the morphological changes observed by AFM." Based on the above considerations, we cited 15-17 in the original manuscript. We acknowledge these three references are not solid enough, so according to your constructive comments, we add more literature to support this key point.

Cite this: *J. Mater. Chem. C*, 2021,
9, 270

Stability of organic thin-film transistors based on ultrathin films of dinaphtho[2,3-*b*:2',3'-*f*]thieno[3,2-*b*]thiophene (DNTT)[†]

An important prerequisite for this investigation is our ability to unambiguously assign the changes observed in the electrical properties of the ultrathin semiconductor films to the mass transport and resulting changes in morphology, as opposed to any other processes, such as any chemical reactions, *e.g.*, the oxidation or decomposition of the molecules during exposure to air and humidity. This makes DNTT the perfect choice for this study, as DNTT is one of the most stable organic semiconductors with respect to long-term exposure to ambient air and elevated temperatures.^{5,23-26} Owing to this excellent chemical stability, the changes in the measured electrical properties of the ultrathin DNTT films can be unequivocally assigned to the morphological changes observed by AFM. The ability to fabricate ultrathin

Fig. R6. Good chemical stability is the prerequisite of the research on aggregate state stability. Ref. 16.

- (2) Several points about Reference 16 are noted here: (a) Reference 16 reported the dewetting phenomenon in thin DNTT films and speculated that the "DNTT monolayer

forms a non-equilibrium phase and is mechanically strained”, which is a kind of support for our work, but **it did not carry out any detailed research on the strain.** (b) Reference 16 also demonstrated that this break-up does not occur when the films have a thickness of more than one monolayer. In fact, the author only deposited 25 nm DNTT film and monitored the morphological evolution **for 12 hours** and electrical performance for **20 days** (Fig. R7). **Our results definitely demonstrate that about 20 nm film is still at the too thin range and suffers from the dewetting problem during the longer storage period. The mobility of 25 nm DNTT OFET decreases by near 20% over a period of 20 days, which further demonstrates that the author didn't resolve the film failure issues.** (c) The authors proposed an encapsulation strategy to improve the stability of thin-film OFETs and only provided the stability data for two months, **which is dramatically shorter than our work and far away from the requirement of practical application.** In contrast, we systematically explore the effect of the intrinsic film strain on the aggregate state stability, proposed a new strain balance strategy, and realized an ultralong lifetime, which is an important advance for improving the stability of OFETs. Therefore, there is an essential difference between Reference 16 and our work.

Cite this: *J. Mater. Chem. C*, 2021, 9, 270

Stability of organic thin-film transistors based on ultrathin films of dinaphtho[2,3-*b*:2',3'-*f*]thieno[3,2-*b*]thiophene (DNTT)[†]

Fig. R7. The morphological evolution and electrical performance of 25 nm DNTT film.

- (3) The statement about strain in reference 17 is fully unsuitable for our case. The authors only considered the strain derived from the structural misfit between the crystalline substrate and the film. There is no lattice misfit in the system consisting of the amorphous substrate and polycrystalline film, so the authors stated that “strain ... caused by the structural misfit ... cannot serve as an explanation for the dewetting in the present case, so that the reason remains unclear”. We recognize there is no

structural misfit-induced strain between the amorphous SiO₂ substrate and DNTT film. However, apart from lattice misfit, there are many other factors such as the thermal expansion mismatch, the molecular conformation-induced stress, or the dislocation among grains that may induce the film strain (Fig. R8).⁴⁻⁸ Besides, according to your suggestions, we further demonstrate the existence of strain in DNTT film by conducting Williamson-Hall (W-H) analysis (for details, see our reply for your comment 5), which is the intrinsic origin of aggregate state instability of organic semiconductor films.

RESEARCH ARTICLE

ADVANCED MATERIALS

Suppressing Interface Strain for Eliminating Double-Slope Behaviors: Towards Ideal Conformable Polymer Field-Effect Transistors

Adv. Mater. 2021, 33, 2101633

As is well-known, the evolution of strain in the semiconductor thin film is tightly linked to the fabrication process.^[23–25] As shown in Figure 1a, when directly depositing the polymer thin film by spin coating, the removal of solvent will enhance the chemical interaction between the film and rigid substrate, thereby constraining the film from expanding or contracting freely, and eventually causing the strain in the film (Figure 1a). After that, the thermal annealing, which is essential to improve film crystallization, can facilitate the further evolution of interfacial interaction. The thermal annealing condition was 150 °C for 60 min. Therefore, the strain in the film is further strengthened owing to the thermal expansion mismatch between the film and rigid substrate (Figure 1aII).^[26–30] To suppress

Figure 1. Strain in the polymer thin film. a–d) Schematic diagram of the film fabrication process and the effect of strain evolution on the molecular structure. e) Out-of-plane XRD of the DPPT-TT thin film before and after thermal annealing. f, g) Out-of-plane and in-plane XRD of the DPPT-TT thin film before and after peeling.

nature materials

REVIEW ARTICLE

<https://doi.org/10.1038/s41569-021-01097-x>

Check for updates

Chem. Mater. 2004, 16, 4497–4508

Pentacene Thin Film Growth

Ricardo Ruiz,^{*,†} Devashish Choudhary,[‡] Bert Nickel,[§] Tullio Toccoli,^{||} Kee-Chul Chang,[†] Alex C. Mayer,[†] Paulette Clancy,[‡] Jack M. Blakely,[†] Randall L. Headrick,[†] Salvatore Iannotta,^{||} and George G. Malliaras^{*,†}

in the second layer than in the first layer. On SiO₂ substrates, for example, the nucleation density is higher for the first layer than for the second layer.³⁰ As a consequence, second layer islands grow on top of more than one first-layer domain. Given that the first layer-islands are randomly oriented with respect to each other, the larger second-layer islands will, in part, be misoriented with some regions of the first layer and this may induce strain. However, the effects of this strain

Origins of strain in halide perovskites

Strain can result from a broad range of factors. During film formation, the precursor solutions are deposited on a substrate and the film is annealed (often above 100 °C). If the film is formed on a substrate with a lower coefficient of thermal expansion (CTE) than the perovskite (~3.3 × 10⁻⁵–8.4 × 10⁻⁵ K⁻¹),¹⁶ the substrate restricts the contraction of the perovskite along the in-plane directions during cool down¹⁷, resulting in an in-plane tensile strain (Fig. 2a, top). Analogously, compressive strain can occur in a film grown on a substrate with a higher CTE than that of the perovskite, though this is less common in halide perovskite devices. This process of cooling after annealing leads to strain (generally tensile) in the plane of the film.

Lattice mismatch between the substrate and the as-formed perovskite film can also generate a tensile or compressive strain (Fig. 2a, middle), depending on whether the *d* spacing of the substrate is larger or smaller than that of the perovskite overlayer. However, when the lattice mismatch is large, the perovskite crystal randomly stacks on the bottom crystal, resulting in a non-uniform interfacial distortion^{18,19}. Compressive and tensile strains can also be introduced to films using mechanical film bending²⁰ (Fig. 2a, bottom) and hydrostatic pressure²¹.

While the above factors generally influence strains across the whole film, other factors, including atomic misfits, phase transitions, light/bias stimulation and grain boundaries, are linked to local strain (namely microstrain) in halide perovskites (Fig. 2b).

Regulating strain in perovskite thin films through charge-transport layers

The correlation between stress (σ) and thermal expansion mismatch is quantified as follows:

$$\sigma_{\Delta T} = \frac{E_p}{1 - \nu_p} (\alpha_s - \alpha_p) \Delta T \quad (1)$$

where E_p is the modulus of the perovskite, ν_p is Poisson's ratio in the perovskite, α_s and α_p are the thermal expansion coefficients of the substrate and the perovskite, respectively, and ΔT is the temperature gradient during cooling from the annealing temperature of the perovskite film to room temperature²². Several

NATURE MATERIALS | VOL 20 | OCTOBER 2021 | 1337-1346 |

16

Fig. R8. The probable origins of strain in organic semiconductor films.

According to the above discussion, aggregate state instability is one of the most important reasons for device failure. Chemical stability is the basic prerequisite of the research on aggregate state instability. Reference 15-17 reported the good chemical stability of DNTT films and the device failure resulting from film dewetting, which indirectly and directly supports our claims regarding the failure of organic transistors. To further consolidate this argument, two other literature are cited here.^{9,10} Thanks for your constructive comments.

Reference

- [1] Henning S. Reliability of organic field-effect transistors. *Adv. Mater.* 38-39, 3859 (2009).
- [2] Jorgensen, M. et al. Stability of Polymer Solar Cells. *Adv. Mater.* 24, 580 (2012).
- [3] Aziz, H. & Popovic, Z.D. Degradation phenomena in small-molecule organic light-emitting devices. *Chem. Mater.* 16, 4522 (2004).
- [4] Wang S. et al. Suppressing Interface Strain for Eliminating Double-Slope Behaviors: Towards Ideal Conformable Polymer Field-Effect Transistors. *Adv. Mater.* 2021, 33, 2101633
- [5] Xue, J. et al. Regulating strain in perovskite thin films through charge-transport layers. *Nat. Commun.* 7, 11156 (2020).
- [6] Wang Z. et al. Revealing molecular conformation-induced stress at embedded interfaces of organic optoelectronic devices by sum frequency generation spectroscopy. *Sci. Adv.* 7, eabf8555 (2021).
- [7] Ruiz, R. et al. Pentacene thin film growth. *Chem. Mater.* 16, 4497 (2004).
- [8] Liu, D. et al. Strain analysis and engineering in halide perovskite photovoltaics. *Nat. Mater.* 20, 1337 (2021).
- [9] Cao, P. et al. Preventing Thin Film Dewetting via Graphene Capping. *Adv. Mater.* 29 (2017).
- [10] Lee, W. H. et al. The Influence of the Solvent Evaporation Rate on the Phase

Separation and Electrical Performances of Soluble Acene-Polymer Blend Semiconductors. *Adv. Funct. Mater.* 22, 267-281 (2012).

Our revision: We modified the statements in the revised manuscript and cited two other publications to support our claim.

Comment 3

The authors claim that organic-semiconductor films with a small thickness often suffer from dewetting caused by tensile strain. The dewetting of thin molecular films is indeed well documented in the literature, but whether this dewetting is caused by mechanical strain or by some other mechanism is not clear at all, as explained in reference 17.

Our reply: As you said, the dewetting of thin molecular films is well documented in the literature, but the underlying mechanism is still unclear now. More importantly, effective suppression of dewetting during long-term storage is not realized in Reference 17. In our work, we demonstrate that strain plays an important role in dewetting and further develop a novel strain balance strategy to effectively suppress the dewetting and to realize an ultra-stable film.

In fact, dewetting is a kind of structural instability. Strain has been well demonstrated to be an intrinsic and basic origin for the structural instability (such as dewetting, phase change, cracks) of the conventional inorganic materials¹⁻³ (such as ferroelectric film, silicon films, and glass coating) and the emerging materials such as perovskite and 2D materials.⁴⁻⁶ In the field of organic transistors, some organic semiconductor films with a small thickness have long been considered to be in a metastable and strained state (Fig. R9), but there is a lack of systematic research on the film strain of organic semiconductor film, its effect on stability, and the strain compensation strategy. Our work well-addressed the above three aspects.

Cite this: J. Mater. Chem. C, 2021,
9, 270**Stability of organic thin-film transistors based on ultrathin films of dinaphtho[2,3-b:2',3'-f]thieno[3,2-b]thiophene (DNNT)†**

polystyrene,³⁰ diindenoperylene³⁶ and pentacene.^{37,38} When a molecular monolayer of, for example, DNNT is deposited onto a solid surface, this monolayer will form during a non-equilibrium phase of the film-growth process and will thus be characterized by significant mechanical strain that results from the competition between adsorption and intermolecular forces.³⁹ If the molecule–substrate interactions are weaker than the intermolecular interactions, thermally activated molecular diffusion will cause this initially closed (or at least connected) molecular monolayer to break up, and a combination of lateral and vertical mass transport will lead to the formation of tall, disconnected islands. This can be considered similar to the

Pentacene Thin Film Growth

Ricardo Ruiz,^{*,†} Devashish Choudhary,[‡] Bert Nickel,[§] Tullio Toccoli,^{||} Kee-Chul Chang,[†] Alex C. Mayer,[†] Paulette Clancy,[‡] Jack M. Blakely,[†] Randall L. Headrick,[⊥] Salvatore Iannotta,^{||} and George G. Malliaras^{*,‡}

indeed, decompose into fragments¹⁰⁰). The calculated structures also suggest that the first monolayer of pentacene adsorbed on the silicon substrate is highly strained and serves as the template for nucleating subsequent layers.

in the second layer than in the first layer. On SiO₂ substrates, for example, the nucleation density is higher for the first layer than for the second layer.³⁰ As a consequence, second layer islands grow on top of more than one first-layer domain. Given that the first layer islands are randomly oriented with respect to each other, the larger second-layer islands will, in part, be misoriented with some regions of the first layer and this may induce strain. However, the effects of this strain have not yet been studied.

Fig. R9. Organic semiconductor films with a small thickness have long been considered to be in a metastable and strained state.

In our work, the effect of strain on the dewetting can be proved from two facts: (1) the existence of strain confirmed by XRD measurements and the related Williamson-Hall (W-H) analysis; (2) the comparative data between the dewetting in the strained film and the stability in strain-free film, both of the films are fabricated and stored under the identical conditions. Except for the thickness, there are no other obvious differences.

As for the first fact, XRD is a typical method to testify the existence of strain, which has been widely utilized in the various thin-film fields including inorganic semiconductors,⁷ organic semiconductors,⁸ perovskites,^{5,6} and 2D materials.⁹ This point is also confirmed by Reviewer #3. As introduced in literature (Fig. R10),⁶ “the aspects of strain in a material that can be determined with typical laboratory techniques include tensile and compressive strain where the interplanar spacing, d , of the material is compared with a reference ‘unstrained’ value. Typically, the variation in d spacing is observed through the Bragg peak positions in an X-ray diffraction (XRD) pattern.” Through systematic measurement of XRD patterns, we found that the thin film and the thick film suffer from tensile strain and compressive strain, respectively. To further confirm the existence of strain and exclude the

influence of polymorphism, we conducted Williamson-Hall (W-H) analysis and peak differentiating and fitting analysis according to your suggestions (for details, see our reply for your comment 5), which firmly demonstrated the existence of strain in DNTT films, based on which we designed an effective strain balance strategy and realized ultra-stable OFETs.

As for the second point, in our work the thin film is prone to dewetting just like some literature, but, interesting, when the thickness is near to a proper value (about 200 nm), film show excellent stability. **In fact, these films are fabricated and stored under the identical conditions. Except for the thickness, there are no other obvious differences.** XRD measurements and W-H analysis shows that they are strained and strain-free film. Therefore, these comparative results as well as the strain-related knowledge and literature strongly prove that the strain is the one important reason for dewetting.

Strain analysis and engineering in halide perovskite photovoltaics

The aspects of strain in a material that can be determined with typical laboratory techniques include tensile and compressive strain where the interplanar spacing, d , of the material is compared with a reference 'unstrained' value. Typically, the variation in d spacing is observed through the Bragg peak positions in an X-ray diffraction (XRD) pattern. If a material is compressively strained, the d spacing is decreased relative to the reference, and the peak associated with that plane will shift to higher scattering vector, q (Fig. 1a,b), with the converse being true for tensile strain. The values of compressive/tensile strain are usually expressed as a ratio of the d spacings and written as percentages. The nature of strain (compressive/tensile) is indicated either explicitly or using a negative/positive sign indicating compressive/tensile strain, respectively. However, we note that

NATURE MATERIALS | VOL 20 | OCTOBER 2021 | 1337-1346 |

Fig. R10. The typical XRD method for strain analysis.

Reference 17 only considered the strain derived from the structural misfit between the crystalline substrate and the film. However, apart from lattice misfit, there are many other

factors such as the thermal expansion mismatch, the molecular conformation-induced stress, or the dislocation among grains that may induce the film strain Fig. R8. **The statements in reference 17 cannot exclude the effect of strain in organic semiconductor films. More importantly, our experimental evidence and related literature prove that strain is one of the most important factors to dewetting.**

Reference

- [1] Borowik, L. et al, The influence of strain on dewetting of silicon films. *J. Appl. Phys.* 114, 063502 (2013).
- [2] Zhang, Y. et al, Effects of Strain and Film Thickness on the Stability of the Rhombohedral Phase of HfO₂, *Phys. Rev. Appl.* 14, 014068 (2020)
- [3] Malzbender, J. & de With, G. Cracking and residual stress in hybrid coatings on float glass. *Thin Solid Films* **359**, 210-214 (2000).
- [4] Dai, Z., Liu, Y., Zhang, Z. Strain Engineering of 2D Materials: Issues and Opportunities at the Interface. *Adv. Mater.* 31, 1805417 (2019).
- [5] Xue, J. et al. Regulating strain in perovskite thin films through charge-transport layers. *Nat. Commun.* 7, 11156 (2020).
- [6] Liu, D. et al. Strain analysis and engineering in halide perovskite photovoltaics. *Nat. Mater.* 20, 1337 (2021).
- [7] Colston, G. et al. Mapping the strain and tilt of a suspended 3C-SiC membrane through micro X-ray diffraction. *Mater. Des.* 103, 244 (2016).
- [8] Wang S. et al. Suppressing Interface Strain for Eliminating Double-Slope Behaviors: Towards Ideal Conformable Polymer Field-Effect Transistors. *Adv. Mater.* 33, 2101633 (2021).
- [9] Wang, M. et al. Enhanced Lithium Storage Performances of Hierarchical Hollow MoS₂ Nanoparticles Assembled from Nanosheets. *ACS Appl. Mater. Interfaces* 2013, 5, 1003–1008

Comment 4

The fact that the dewetting of monolayer-thick DNTT films can be alleviated by increasing the thickness of the DNTT films has been reported previously (reference 16).

Our reply: There is an essential difference between Reference 16 and our work.

(1) Although reference 16 reported the thicker (25 nm) film is more stable than thinner (2.5 nm) film, our results definitely demonstrate that about 20 nm film is still in the too thin range and suffers from the dewetting problem during the longer storage period, **because the strain in the 20 nm DNTT film was not balanced and thus the dewetting problem cannot be completely alleviated (Fig. 2a, 2b, and Supplementary Fig. 2)**. As shown in Fig. R11, even 40 nm DNTT film still suffers from dewetting, **which demonstrates that the simple increasing thickness method without the guidance of strain balance principle cannot fundamentally solve the instability problem of DNTT films.**

Exceptional Dewetting of Organic Semiconductor Films: The Case of Dinaphthothienothiophene (DNTT) at Dielectric Interfaces

Figure 5. (a) Morphologies as determined from AFM micrographs of 2 nm thick DNTT films after different aging times. (b) Corresponding quantitative evaluation of surface covering and average island height. (c) Dewetting of DNTT multilayer film ($d_{\text{nom}} = 40$ nm). Despite the larger time scales, again a distinct structural evolution is observed. (We note that the used z -scales in panel c are nonequal to account for the modified vertical extension of the films after different times of aging. To allow for a correlation with the actual heights, corresponding line scans are provided in Figure S3.)

Fig. R11. Dewetting of 40 nm DNTT film.

- (2) **The mobility of the 25 nm DNTT OFET decreases by near 20% over a period of 20 days, which further indicates that the author didn't resolve the film failure issues.**
- (3) Reference 16 proposed an encapsulation strategy to improve the stability of thin-film OFETs and only provided stability data for two months (with mobility degradation of 20%). **It is far away from the requirement of practical application.**

Therefore, compared to reference 16, our work provides a novel and deep understanding as well as novel and efficient solution to device instability, which guarantee the novelty and significance of our work.

Section “Thickness-dependent strain in OSC film”:

Comment 5

The analysis of the mechanical strain is rudimentary and likely flawed. The authors have only calculated (using Equation 1) the relative difference between the (001) d-spacing of the DNTT films (measured by x-ray diffraction) and that of a DNTT crystal and then assumed that this relative difference in d-spacings represents the mechanical strain in the DNTT films. However, the observed d-spacings are more likely the result of polymorphism, rather than strain, since vacuum-deposited films of small-molecule semiconductors, such as DNTT, are often characterized by a coexistence of various phases with different d-spacings (Appl. Phys. Lett., vol. 90, p. 081903, 2007; Phys. Rev. B, vol. 85, p. 195308, 2012; Adv. Funct. Mater., vol. 26, p. 2233, 2016; Adv. Mater., vol. 29, p. 1604833, 2017; ACS Appl. Mater. Interfaces, vol. 9, p. 11977, 2017). Since the authors have not performed a Williamson-Hall (W-H) analysis, their claims regarding the magnitude of the strain in the DNTT films are purely speculative.

Our reply: Thanks for your valuable suggestions.

(1) Our reply for comments 3 shows that XRD is a typical method to testify the strain in various materials, and that the change of XRD peaks indicates the existence of strain, but it does not take polymorphism into considerations. Based on your opinion, X-ray diffraction (XRD) analysis was carried out with a peak differentiating and fitting method to separate the thin-film phase (FP) and the bulk phase (BP). Because the difference of d-spacing between the thin-film phase and the bulk phase of DNTT is very small, the (001) diffraction peak of DNTT film in original Fig. 1b is very symmetrical and cannot reflect the polymorphism. According to your suggestion, we further measured and analyzed the diffraction peaks of high orders. At high order peaks, the polymorphism appears from the splitting diffraction peaks. Taking 300 nm DNTT film as an example, the diffraction peaks of (003), (004), and (005) crystal planes are asymmetrical, especially in the (005) crystal plane, which has been completely spitted into two

separated peaks (Fig. R12). However, for relatively thin films, the intensity of the higher-order peaks after the (003) peak is very weak or even undetectable. Therefore, we finally chose the diffraction peaks of the (003) crystal plane to carry out peak differentiating and fitting, and polymorphism analysis of DNTT films with different thicknesses. **Consequently, we individually analyzed the separated thin-film phase (FP) and bulk phase (BP) and found that the diffraction peaks of both phases shift to a larger diffraction angle with the increase of the thickness (Fig. R13).** The corresponding d -spacing of the thin-film phase increases from 1.6294 nm to 1.6345 nm and the bulk phase increases from 1.6178 nm to 1.6243 nm. **These results solidly confirm the existence of strain in DNTT films.**

Fig. R12. The XRD patterns of the 300 nm DNTT film and its peak differentiating and fitting

results of the (003), (004), and (005) crystal planes.

Fig. R13. The peak differentiating and fitting results of the (003) crystal planes of DNTT films with different thicknesses.

- (2) According to your suggestion, under the assistance of XRD experts (Dr. Haiyan Du and Dr. Xiaoxia Hu who are in charge of XRD equipment in the Analysis and Testing Center of Tianjin University) and strain expert (Prof. Langli Luo who is skilled in strain characterization and analysis, Chem. Commun. 56, 13301 (2020); Nat. Commun. 9, 2437(2018); Nat. Commun. 9, 2924(2018)), **we performed Williamson-Hall (W-H) analysis on DNTT films with different thicknesses and calculated the strain after excluding the effect of instrument-induced broadening. The analysis results (Fig. R14) clearly show that organic semiconductor films suffer from strain.**

Below are the details of W-H analysis:

In the W-H method, it is assumed that the peak broadening in X-ray diffraction data is mainly due to the contribution of grain size and strain after excluding the instrument broadening.

Therefore, the broadening of the peaks can be expressed as

$$\beta_T = \beta_D + \beta_\varepsilon \quad (1)$$

Where β_T is the total broadening, β_D is the broadening caused by crystallite size, β_ε is the broadening caused by the strain.

The broadening of the crystallite size can be given by the Scheler equation.

$$D = K\lambda / \beta_D \cos \theta \quad (2)$$

where D is the grain size, β_D is the full width at half maxima (FWHM) of diffraction peak, and K is the shape factor ($K = 0.94$). λ is the wavelength of the X-ray source ($\lambda = 0.15405$ nm), and θ is the position of the diffraction peak.

The X-ray diffraction broadening caused by strain is given by Wilson's formula,

$$\beta_\varepsilon = 4\varepsilon \tan \theta \quad (3)$$

where β_ε the X-ray diffraction broadening caused by the strain, ε is the strain. Thus, combining equations (2) and (3) in equation (1), the W-H equation is usually written as:

$$\beta_T = K\lambda / D \cos \theta + 4\varepsilon \tan \theta \quad (4)$$

It also can be written as:

$$\beta_T \cos \theta = K\lambda / D + 4\varepsilon \sin \theta \quad (5)$$

The strain ε can be estimated from the slope of the $(\beta_T \cos \theta)$ vs. $(4 \sin \theta)$ graph (the Williamson-Hall plot).

Fig. R14. The strain in DNTT films with different thicknesses by Williamson-Hall (W-H) analysis.

Our revision: The detailed strain analysis by XRD is added in the revised manuscript (Page 6) and Supplementary Information (Section 1).

Comment 6

Besides, if there was indeed significant strain in the DNTT films, it is highly unlikely that the strain would display a monotonic dependence on the film thickness (as the authors indicate in Fig. 1d); it is far more likely that a transition from a surface layer to a bulk layer would occur beyond a critical film thickness (Adv. Funct. Mater., vol. 26, p. 2233, 2016).

Our reply: Our reply to comments 5 (mainly referring to the XRD and W-H analysis)

strongly confirm that there is indeed significant strain in the DNTT films. As shown in Fig. R13, the diffraction peaks of both the thin-film phase and the bulk phase shift monotonically with the film thickness. This proves that the *d*-spacing shift is not only due to the transition from a surface layer to a bulk layer but also from the film strain.

If the diffraction peak shift is only caused by the phase transformation, the diffraction peaks of the thin-film phase would shift to the location of the bulk phase when the film thickness reaches a critical value, while the bulk phase diffraction peaks would not change, but this is not the case. We individually analyzed the separated thin-film phase (FP) and bulk phase (BP) and found that the diffraction peaks of both phases shift to a larger diffraction angle with the increase of the thickness (Fig. R13). The corresponding *d*-spacing of the thin-film phase increases from 1.6294 nm to 1.6345 nm and the bulk phase increases from 1.6178 nm to 1.6243 nm. This suffices to prove that there is strain in DNTT films with different film thicknesses.

Comment 7

The authors report that they did not observe any changes in the morphology of the nominally 200-nm-thick DNTT films over a period of five years using atomic force microscopy (Fig. 2c,d), but this AFM analysis is meaningless, since AFM images show only the surface of the films, but not the morphology of the material buried underneath the surface.

Our reply: This comment points out the basic limitation of AFM measurement, but it does not deteriorate the key arguments of this work. As you said, AFM images show only the surface of the films and is not possible to inspect the morphology of the material buried underneath the surface, but **the surface morphology is an important indicator for the film structure**. In our case, the possibility of a stable surface but an unstable interface can be excluded by XRD and electrical measurement. **Especially the stable electrical property indicates that there should be no significant change at the buried interface (conductive channel) because charge transport is highly sensitive to structural**

variation.

In addition, AFM is a widely used measurement method to monitor the morphological evolution of organic films (Nat. Energy, 5, 947 (2020), ACS Appl. Mater. Interfaces 9, 8384(2017), ACS Nano 15, 1155(2021)) in OFET, OLED, and OPV, in which the photoelectronic process mainly occurs in the buried interface rather than the surface (Fig. R15). In fact, AFM measurements have provided very meaningful information for understanding the photoelectronic properties and promoted the development of organic electronics and other fields with thick active film. Therefore, we should rationally treat the possible difference between surface and bulk. **If this point is treated too critically, most published work mainly based on AFM measurements should not be published or be retracted.** Certainly, the combination of AFM with other techniques to characterize the surface and bulk simultaneously would be the best way, which has been performed to some extent in this work.

Fig. R15. The widely used AFM method in the literature.

Section "Interpretation of the thickness-dependent strain and aggregate state stability":

Comment 8

The authors claim that the lattice parameters of the DNTT films are different from the lattice parameter of the amorphous silicon dioxide gate dielectric. This makes no sense, because

amorphous materials (such as silicon dioxide) obviously do not have any lattice parameters, which also means that there is no misfit and hence no misfit strain.

Our reply: Thanks for your precious comments and suggestions. Indeed, amorphous materials (such as silicon dioxide) do not have lattice parameters. Actually, the inaccuracy statement had been revised in the manuscript.

Comment 9

The authors claim that the thermal-expansion coefficient of DNTT is different from that of the silicon dioxide. This statement is meaningless, since the thermal-expansion coefficients of single-crystalline and polycrystalline organic semiconductors are usually highly anisotropic, and the thermal-expansion coefficient often has a positive value in one lattice direction, but a negative value in another lattice direction.

Our reply: We are sorry that we cannot agree with your statement about this item. Firstly, **the previous works have shown that the different thermal expansion coefficients of organic semiconductors and inorganic substrate can cause a notable interface strain** (Appl. Phys. Lett. 110, 141904 (2017), ACS Appl. Mater. Interfaces 12, 40607(2020), Nat. Commun. 3, 723 (2012), Nat. Commun. 7, 10270 (2016), PNAS 114(33), E6739(2017)), as shown in Fig. R16. Secondly, you say that “the thermal-expansion coefficients of single-crystalline and polycrystalline organic semiconductors are usually highly anisotropic, and the thermal-expansion coefficient often has a positive value in one lattice direction, but a negative value in another lattice direction”. This statement is not true. **Organic molecules stack together by weak intermolecular interactions. The positive thermal expansion (PTE) is a usual phenomenon in organic materials, and the negative thermal expansion (NTE) is extremely rare** (Nat. Mater. 9, 36(2010), J. Phys. Chem. C 124, 27413(2020), Chem. Commun. 54, 10675(2018), Acta Cryst. B 77, 309(2021)), as shown in Fig. R17. **Moreover, even there is PTE and NTE in anisotropic organic materials, it is an extremely low possibility that PTE and NTE are the same.** Therefore, the overall effect of the difference of thermal expansion coefficient between organic materials and inorganic substrates cannot be ignored (Fig. R18. IEEE T. Electron

Dev. 59(1), 225(2011)).

Fig. R16 Interface strain stemmed from the different thermal-expansion coefficients of organic semiconductors and inorganic substrate.

Fig. R17 The positive thermal expansion (PTE) is a usual phenomenon in organic materials, and the negative thermal expansion (NTE) is extremely rare.

IEEE TRANSACTIONS ON ELECTRON DEVICES, VOL. 59, NO. 1, JANUARY 2012

Thermal Expansion Coefficient Considerations on Field-Effect Mobility of Pentacene Organic Thin-Film Transistors With an AlN Gate Dielectric

Most surface treatments have focused on the chemical, rather than physical, properties at the pentacene/dielectric interface. The interfacial strain is one of the physical properties, which may be induced due to the mismatch of the coefficient of thermal expansion (CTE). However, interfacial stress is usually ignored for organic thin-film devices owing to the relatively soft properties. In this article, we present a simple method to im-
contraction in our case. First, during deposition, the pentacene thin-film phase on AlN is considered to be stress free at the chosen substrate temperature. Second, when the pentacene thin-film phase is cooled down to RT, a tensile stress is generated in plane in the pentacene thin-film phase due to the difference in the CTE between pentacene and AlN. Note that there exists a negative CTE along the a -axis ($\sim -10^{-5} \text{ K}^{-1}$) and a large positive CTE along the b -axis ($\sim 10^{-4} \text{ K}^{-1}$) in the single crystal of pentacene [24]. To first-order approximation, it is reasonable to assume that the overall effect of CTE is positive in plane for simplicity. The induced tensile stress in plane in the pentacene film is thus attributed to the CTE of pentacene larger than AlN ($4 \sim 5 \times 10^{-6} \text{ K}^{-1}$ [25]), which may cause the c -axis

Fig. R18. the overall effect of the difference of thermal expansion coefficient between organic materials and inorganic substrates cannot be ignored.

Section "Effect of thickness-dependent strain on OFET stability":

Comment 10

In Table 1, the authors ignore all previous publications in which the excellent long-term stability of DNTT transistors has been reported (reference 15; Org. Electronics, vol. 15, p. 1998, 2014; Adv. Electron. Mater., vol. 2, p. 1500209, 2016; Org. Electronics, vol. 61, p. 65,

2018).

Our reply: Thanks for your reminder. We cited 8 papers about the long-term stability of OEFTs in Table 1, but only one paper is based on DNTT OFET. The papers you mentioned (reference 15; Org. Electronics, vol. 15, p. 1998, 2014; Adv. Electron. Mater., vol. 2, p. 1500209, 2016; Org. Electronics, vol. 61, p. 65, 2018) are all about the stability of DNTT OEFTs, but the stability-improvement method and its mechanism are totally different from our work. More importantly, the stability performance (lifetime and mobility degradation percent) in those references are significantly inferior to our devices.

In reference 15, the electrical stability of bottom-gate, top-contact DNTT OTFTs fabricated on a polystyrene-buffered, flash-evaporated TPGDA dielectric has been investigated and hole mobility was stable over the **9 months**. In (Org. Electronics, vol. 15, p. 1998, 2014), the DNTT OFETs with a vacuum-evaporated polymer dielectric layer (TPGDA) are stable with about 8.3% degradation after storage in air for **eight months**. Although Reference 15 and (Org. Electronics, vol. 15, p. 1998, 2014) claim that their devices show good (or excellent) stability, such lifetime is much shorter than our devices and far away from the requirements (several years lifetime) of practical applications. If reference 15 really achieved excellent stability in 2014 as it claimed, there should be not stability problem in organic transistors, and organic transistors might be promoted toward applications in early time.

In (Adv. Electron. Mater., vol. 2, p. 1500209, 2016), the mobility of DNTT OFETs with three kinds of ultrathin polymer dielectrics decreases by about **38%, 80%, and 90%** during **140 days storage**.

In (Org. Electronics, vol. 61, p. 65, 2018), a relatively longer shelf-life of **470 days** but larger degradation of **20.8%** was obtained by using stencil lithography based on high-resolution silicon stencil masks to fabricate DNTT OTFTs with a channel length of 0.3 μm . Such devices could not be treated as stable devices.

Therefore, it can be found that the strategies in the mentioned papers are different from our strain balance strategy, and the shelf-life is much shorter than the 5 years realized in our work.

Our revision: We have cited the related literature in Table 1 (Page 14).

Section “Self-optimized contact resistance by the spontaneous penetration of gold nanoclusters”:

Comment 11

The contact resistance of the DNTT transistors reported here (even for a nominal DNTT thickness of 20 nm) is about an order of magnitude larger than what has been reported in the literature for DNTT transistors; this indicates that there are problems with the device fabrication process employed here.

Our reply: The lowest value of the contact resistance of the 20 nm DNTT transistor is 1.83 k Ω ·cm (Supplementary Fig. 13), which is at the same level as that of DNTT transistors with the same configuration in the literature (ACS Nano 15, 1155(2021) , Electron. Mater. Lett. 15, 166(2019)), as presented in Fig. R19. We acknowledge that the contact resistance of our devices is still higher than some values that have been achieved by the elaborate electrode/semiconductor interface engineering. However, **our contact resistance is only obtained by the natural gold/semiconductor contact without any optimization. We believe that the high contact resistance in our device can be further reduced by interface engineering.**

Blurred Electrode for Low Contact Resistance in Coplanar Organic Transistors

TC configuration, respectively. With the increasing V_{OV} , the extracted R_c ranges from 5.01 to 2.32 k Ω cm for BC geometry (adjusted R^2 values >0.98), which is comparable to the TC geometry (from 3.25 to 2.17 k Ω cm). In addition, we calculate

Electronic Materials Letters (2019) 15:166–170
<https://doi.org/10.1007/s13391-018-00112-9>

ORIGINAL ARTICLE - ELECTRONICS, MAGNETICS AND PHOTONICS

Improved Device Ideality in Aged Organic Transistors

Chang-Hyun Kim¹ 

Fig. 3 **a** Illustration of the distribution of the applied potential V_D along the transistor channel. **b** Extracted V_G -dependent V_c and R_c in a pristine OFET (inset: statistical analysis of the resistance ratio R_c/R_{tot} as a function of V_G)

Fig. R19. The contact resistance in the literature

The electrical parameters (mobility, on/off ratio, threshold voltage, contact resistance)

of our devices are all in the normal range, which guarantees the fabrication process. In fact, I started to work in the field of organic transistors since my Ph.D. program in 2005 (Adv. Mater. 19, 2613(2007); Adv. Mater. 19, 2613(2007); J. Am. Chem. Soc. 132, 8807(2010); Adv. Energy Mater. 1, 188(2011); Adv. Mater. 24, 2159(2012); Adv. Mater. 24, 3053(2012); Adv. Mater. 25, 3419(2013); Angew. Chem. Int. Ed. 52, 12530(2013)), and continue to work in this field now (Adv. Mater. 31, 1805630(2019); Nat. Commun. 12, 21(2021); Sci. Adv. 7, eabf8555(2021); Adv. Mater. 2104166(2021)). Therefore, I have confidence in our fabrication process. In fact, I pay attention to the instability problem of organic transistors since my Ph.D. program but continued to get minor progress during the past ten years. In the beginning, we thought the organic semiconductor is unstable, so we use ultra-stable titanyl phthalocyanine to fabricate the device, but it also show degradation. This fact indicates that the chemical factor is not the key to device instability. Then we continued to dig out the underlying reasons year by year. Until this work, we finally find out the key role of stain and develop an efficient solution.

Comment 12

The observation that the contact resistance of organic TFTs decreases over time has also been reported (see, for example, Electronic Mater. Lett., vol. 15, p. 166, 2019; Phys. Rev. B, vol. 76, p. 184112, 2007).

The fact that this improvement in contact resistance may be related to the penetration of gold clusters into the semiconductor film has also been known for a long time (see, for example, Adv. Mater., vol. 14, p. 961, 2002; J. Appl. Phys., vol. 93, p. 5201, 2003; Appl. Phys. A, vol. 95, p. 225, 2009; Org. Electronics, vol. 15, p. 1738, 2014; Org. Electronics, vol. 69, p. 92, 2019; Jpn. J. Appl. Phys., vol. 58, p. S11B28, 2019; ACS Appl. Mater. Interfaces, vol. 12, p. 30616, 2020).

Our reply: Thank you for recommending these relevant literatures. However, these literatures cannot demonstrate our work lack novelty.

Firstly, we want to note that the contact resistance evolution due to the gold penetration is technical, but not the key point of this work, which just proves thick film would not suffer

from the large contact resistance. The **key novelty of this work** is as follows: it is the first time to reveal that the OFET instability stemmed from the residual strain in OSC film, and further reported a novel and efficient strategy to balance the strain and significantly improve the device stability, which is based on the unique transition from tensile strain to compressive strain with the increase of OSC film thickness. The tensile strain in the too-thin film and the compressive strain in the too-thick film lead to dewetting and crack problems, respectively. Moreover, an ultra-stable organic transistor for 5 years is realized with the strain balance strategy, which has never been reported before.

In ref. (Electronic Mater. Lett., vol. 15, p. 166, 2019 and Phys. Rev. B, vol. 76, p. 184112, 2007), the authors have reported that the contact resistance of organic TFTs decreases over time. **However, the former explored the origin of ideality improvement in aged DNTT transistors, and the latter reported on the healing of defects at room temperature in the organic semiconductor pentacene. They are totally different from our work.**

Reply for Reviewer 3:

Reviewer #3 (Remarks to the Author):

Comment 1

It is well-known that stability is one of the most important and challenging issues in the field of organic electronics. In “Balancing the Film Strain of Organic Semiconductor for Ultra-stable Organic Transistors with a Five Years Lifetime”, the authors report a stable organic field-effect transistor (OFET) with 5 years of lifetime by a strain balance strategy, and disclose that the intrinsic tensile and compressive strain existing in the organic semiconductor (OSC) film are the key origins of the aggregate state instability and device degradation. To the best of my knowledge, this truly impressive 5-year lifetime is a record result. The strain balance strategy based on thickness-dependent strain evolution in organic semiconductors opens up a new perspective to device stability in organic electronics. The results resolve one of the key challenges in the field of organic electronics in a unique and new way. The topic of this work is important and would be appealing for the broad readers of Nature Communications from both academia and industry. Therefore, I recommend publishing this manuscript in Nature Communications after minor revisions.

Our reply: Thank you very much for your positive and insightful comments and for your recognition of the significance of our work! We will answer your questions one by one below.

Comment 2

XRD is a typical method for strain analysis. The authors are suggested to discuss the method and the representative literature (Nat. Mater. 20, 1337 (2021); Nat. Mater. 20, 1337 (2021)).

Our reply: Thanks for your constructive suggestion. For crystalline materials, strain is inevitable and can be from external forces and internal structural defects such as atomic interstitials, dislocation, and vacancies. Therefore, the lattice parameters of crystalline

materials are variable due to the existence of strain. For single-crystal materials, the lattice parameters are constant and present strain-free due to the periodically arranged atoms/molecules. Taking the strain-free single-crystal material as a reference, the strain value of crystalline materials can be determined as a tensile or compressive state by measuring the lattice parameters. As a typical and high-precision laboratory technique, XRD has a great advantage to characterize the micro- and nano-scale deformation of crystalline materials. The variation in d -spacing is a good parameter to quantitatively describe the lattice deformation and can be easily extracted from the Bragg diffraction peaks of XRD patterns, which is widely used in organic semiconductors, perovskites, and two-dimensional materials.

Our revision: We have added the discussion about XRD and strain analysis and cited the representative literature in the revised manuscript (page 5).

Comment 3

In Fig. 2b, the film became discontinuous after storage. Where are the disappeared semiconductor molecules? The authors are suggested to explain this phenomenon.

Our reply: Fig. 2b (Fig. R20 here) shows the atomic force microscope (AFM) images and section profiles of the 20 nm DNTT film before and after five-year storage. We can see that the 20 nm DNTT film underwent a significant dewetting process during the long-term storage, which is reported in previous work (Nat. Commun. 10, 3872 (2019)). The molecules at the grain boundary, especially in the thinner films, could diffuse to the adjacent grains or upper layer of the same grain, which leads to the increase of height of the crystal grains of the 20 nm DNTT film (Fig. R20), and thus many holes and discontinuous networks formed during the dewetting process.

Our revision: The corresponding analysis is discussed in maintext (page 8).

Fig. R20. Atomic force microscope images and section profiles of the 20 nm DNTT films before and after five-year storage.

Comment 4

In lines 140-142 on page 8, the authors state that “it is reported that the growth behavior in ultrathin films is mainly determined by the substrate properties such as the lattice parameter misfit and the mismatch of coefficient of thermal expansion”. I think the

mismatch of surface energy should also be taken into account.

Our reply: Thanks for your professional suggestion. The mismatch of surface energy is indeed one of the main factors to influence the growth behavior in ultrathin films. It is reported that the phase of organic semiconductors is influenced by the surface energy of the substrate and crystal plane. As shown in Fig. R21, the free energy of semiconductor film is synergistically determined by the energy of substrate and semiconductor crystals. Organic semiconductor molecules are prone to a stacking mode with low free energy. Therefore, the mismatch of surface energy has an important effect on the growth behavior of ultrathin films.

Adv. Mater. **2005**, *17*, No. 7, April 4

We have developed a thermodynamic model to explain the preference for growth of the orthorhombic thin-film phase crystal structure at low film thickness. For a crystal with sides of length l and height h , the energy E can be written as:

$$E = l^2\gamma_{001} + l^2\gamma_s + 4hl\gamma_{hk0} + hl^2b \quad (1)$$

where γ_{001} and γ_{hk0} are the surface energies of the (001) and (hk0) type planes, respectively, γ_s is the interfacial energy between the crystal (001) plane and the substrate, and b is the bulk volume energy of the crystal. The detailed XRD studies

JOURNAL OF APPLIED PHYSICS **101**, 033522 (2007)

$T_g=90$ °C.⁹ The unbuffered interface is structurally unstable—the ITO surface is hydrophilic while TPD and NPB are hydrophobic, resulting in a large surface energy mismatch and interfacial instability. Several studies have

Fig. R21. The thermodynamic growth model of organic semiconductor molecules.

Our revision: The analysis about the mismatch of surface energy is added in the maintext (page 9) and the corresponding citation is cited in references 46 and 47.

Comment 5

In lines 218-219 on Page 12, “the 200 nm DNTT device exhibits a much longer lifetime and

218 remains high performance up to 90% even after five-year storage". It seems that the mobility after five-year storage is even slightly higher than that of the fresh device, as presented in Fig. 3b. Please explain it.

Our reply: Sorry for the unclear expression. Actually, the mobility after five-year storage is even slightly higher than that of the fresh device, as presented in Fig. 3b. The mobility is influenced by the transport at the semiconductor/dielectric interface and injection at the semiconductor/electrode interface in OFET. In the 200 nm DNNT device, the injection barrier (i.e. contact resistance) at the semiconductor/electrode interface decreases largely at the first month of storage, which leads to the increase of the mobility and the highest mobility after one month of storage. To show the degradation of the device mobility more objectively, the mobility of the 200 nm DNNT device after five-year storage is compared to that at the first month of storage, corresponding to the highest mobility of the 200 nm DNNT device.

Our revision: The calculation method and the origin for the increase of mobility in the first month is briefly described in Supplementary Information (Section 4).

Comment 6

Supplementary Fig. 10 is not referenced in the manuscript. Please check and correct it.

Our reply: Thanks for your kindly remind. Supplementary Fig. 10 in the original manuscript shows the gate voltage-dependent mobility of the 200 nm DNNT OFET before and after one-month storage. It is used to prove the accuracy of the extracted mobility, which is important to evaluate the evolution of mobility.

Our revision: The corresponding discussion is added in the revised manuscript (page 16).

Reviewer comments, second review

Reviewer #1 (Remarks to the Author):

The practical application of organic transistors has been desired. There has been, to my knowledge, no report about 5 years stability tests as reported in this manuscript. The authors finding that the film thickness is the key to achieve high stability will be useful for device researchers. All my questions and comments were appropriately answered and the revised manuscript is now suitable for publication.

Reviewer #2 (Remarks to the Author):

In the revised manuscript, the authors have addressed my previous comments, so I recommend publication in Nature Communications.

Reviewer #3 (Remarks to the Author):

I have gone through the entire manuscript. The authors did a very careful revision of the manuscript, and adequately addressed the questions/comments I raised earlier.

I also went through the responses to the questions and comments raised by the other two reviewers. I think the authors gave a professional and detailed response, and well addressed those questions and comments.

As such, I would recommend accepting this manuscript by Nature Communications.

Reviewer #1 (Remarks to the Author):

The practical application of organic transistors has been desired. There has been, to my knowledge, no report about 5 years stability tests as reported in this manuscript. The authors finding that the film thickness is the key to achieve high stability will be useful for device researchers. All my questions and comments were appropriately answered and the revised manuscript is now suitable for publication.

Our reply: We deeply appreciate your approval of the significance of our work. Your professional and constructive comments and suggestions guide us to think about some points more deeply and are very helpful to improve the quality of the manuscript.

Reviewer #2 (Remarks to the Author):

In the revised manuscript, the authors have addressed my previous comments, so I recommend publication in Nature Communications.

Our reply: We highly appreciate you for your critical and constructive comments, because your comments truly guide us to think about some points deeply and improve our manuscript.

Reviewer #3 (Remarks to the Author):

I have gone through the entire manuscript. The authors did a very careful revision of the manuscript, and adequately addressed the questions/comments I raised earlier.

I also went through the responses to the questions and comments raised by the other two reviewers. I think the authors gave a professional and detailed response, and well addressed those questions and comments.

As such, I would recommend accepting this manuscript by Nature Communications.

Our reply: Thank you very much for your positive and insightful comments and for your recognition of the significance of our work!